# Catalytic production of impurity-free $V^{3.5+}$ electrolyte for vanadium redox flow batteries

Jiyun Heo[1], Jae-Yun Han[2], Soohyun Kim[1], Seongmin Yuk[1], Chanyong Choi[1], Riyul Kim[1], Ju-Hyuk Lee[1], Andy Klassen[3], Shin-Kun Ryi[2]* & Hee-Tak Kim [1,4]*

The vanadium redox flow battery is considered one of the most promising candidates for use in large-scale energy storage systems. However, its commercialization has been hindered due to the high manufacturing cost of the vanadium electrolyte, which is currently prepared using a costly electrolysis method with limited productivity. In this work, we present a simpler method for chemical production of impurity-free $V^{3.5+}$ electrolyte by utilizing formic acid as a reducing agent and Pt/C as a catalyst. With the catalytic reduction of $V^{4+}$ electrolyte, a high quality $V^{3.5+}$ electrolyte was successfully produced and excellent cell performance was achieved. Based on the result, a prototype catalytic reactor employing Pt/C-decorated carbon felt was designed, and high-speed, continuous production of $V^{3.5+}$ electrolyte in this manner was demonstrated with the reactor. This invention offers a simple but practical strategy to reduce the production cost of $V^{3.5+}$ electrolyte while retaining quality that is adequate for high-performance operations.

[1] Department of Chemical and Biomolecular Engineering, Korea Advanced Institute of Science and Technology, 291, Daehak-ro, Yuseong-gu, Daejeon, Republic of Korea. [2] Advanced Materials and Devices Laboratory, Korea Institute of Energy Research (KIER), 152 Gajeong-ro, Yuseong-gu, Daejeon, Republic of Korea. [3] Avalon Battery, 3070 Osgood Ct Fremont, Fremont, CA 94539, USA. [4] Advanced Battery Center, KAIST Institute for the NanoCentury, KAIST, 291, Daehak-ro, Yuseong-gu, Daejeon, Republic of Korea. *email: h2membrane@kier.re.kr; heetak.kim@kaist.ac.kr

Environmental concerns are driving the development and use of renewable energy sources with better quality and greater quantity[1,2]. However, due to the unsteady nature of renewable energy sources, electrochemical energy storage (EES) is in high demand to make practical use of renewable energy in grid applications[3–5]. The redox flow battery is a very promising system among the candidates for the use in EES due to its large energy capability, high safety, and flexible control of the energy-to-power ratio[6,7]. Notably, an all-vanadium redox flow battery (VRFB) has received much attention because it has high efficiency and long life time without concern about cross-contamination[8,9]. Yet, the commercialization of VRFB is still hindered due to the expensive cell components despite such advantages[10]. In particular, vanadium electrolyte accounts for a large portion of VRFB costs because of the need for expensive vanadium precursor materials and the high cost of electrolyte production. For example, for systems of 10 kW/120 kWh, the cost for vanadium and electrolyte production cost account for 40 and 41%, respectively, of the total energy cost[11]. Furthermore, the portion of the electrolyte cost in the total VRFB cost increases with increasing energy capacity of a system[12,13]. Therefore, cost-effective production of VRFB electrolyte must be developed to achieve broader acceptance of VRFB[13,14].

The $V^{3.5+}$ electrolyte, which is an equimolar mixture of $V^{4+}$ ($VO^{2+}$) and $V^{3+}$ electrolyte, is especially preferred in industry as both positive and negative electrolytes because VRFB can be operated without initial re-balancing of its positive and negative capacity. A full charging of VRFB with the use of the same $V^{3.5+}$ electrolyte for positive and negative electrodes results in $V^{5+}$ and $V^{2+}$ electrolyte at the positive and negative electrodes, respectively. In most cases, $V_2O_5$ is commonly used as a vanadium source for preparing $V^{3.5+}$ electrolyte because of its low cost, compared with other vanadium precursors. The conventional route for preparing $V^{3.5+}$ electrolyte from $V_2O_5$ includes the chemical reduction of $V^{5+}$ ($VO_2^+$) to $V^{4+}$ with a reducing agent and the electrolysis of $V^{4+}$ electrolyte to produce $V^{3+}$ electrolyte[15]. The reduction of $V^{5+}$ to $V^{4+}$ can easily be achieved with a residue-free organic reducing agent (ORA) such as oxalic acid[16–18]. However, the reduction of $V^{4+}$ to $V^{3+}$ with ORA is quite sluggish, which presents a major challenge for achieving practical chemical production of $V^{3.5+}$ electrolyte. Therefore, instead of chemical reduction, electrolysis of $V^{4+}$ electrolyte has been employed using a VRFB stack[17,19,20]. As indicated by the inventions from Skyllas–Kazacos's group, the reduction of $V^{4+}$ electrolyte at the negative electrode can be coupled with either oxidation of $V^{4+}$ at the positive electrode or water splitting reaction[16,21]. Preparing $V^{3.5+}$ electrolyte by using $V^{4+}$ electrolyte in the positive electrode during electrolysis ensures impurity-free production, but an additional reduction process for the surplus $V^{4.5+}$ from the anode is required. On the other hand, the electrolysis method based on water splitting reaction can prevent the generation of surplus $V^{4.5+}$ electrolyte. However, advanced engineering may be needed to address the vanadium ion crossover to oxygen evolution reaction (OER) electrode and the carbon corrosion at OER electrode[22].

In search of simpler production, a few chemical processes have been attempted. The use of $V_2O_3$ enables the production of impurity-free $V^{3+}$ electrolyte;[18] however, the slow dissolution rate and high cost of $V_2O_3$ restrict its application. Tanaka et al.[23] invented a method by which to prepare a $V^{3+}$ electrolyte by mixing $V_2O_5$ and sulfur, followed by calcination process to form a soluble $V^{3+}$ compound. Although this method enables the chemical production of $V^{3.5+}$ electrolyte from $V_2O_5$, its complexity, high temperature processing conditions (200–300 °C), and the possibility of toxic $SO_2$ gas generation inhibit its application. Aside from the ORAs, alkali or transition metals with low redox

potential, which are strong reducing agents, can reduce $V^{4+}$ to $V^{3+}$, although with release of some ion impurities[24]. However, the ions negatively influence VRFB performance by metal deposition and consequently accelerate hydrogen evolution[13]. For these reasons, inventing a new greener and simpler method to replace electrolysis when producing impurity-free vanadium electrolyte promises to offer a significant advance in VRFB technology.

Here, we present a large-scalable method for producing impurity-free $V^{3.5+}$ electrolyte using catalyzed oxidation of ORAs. The use of ORAs is beneficial for obtaining impurity-free electrolyte because only water and easily removable $CO_2$ are released through the reaction; however, the chemical reduction of $V^{4+}$ to $V^{3+}$ by the oxidation of ORAs remains unachieved due to the low reactivity of ORAs. To accelerate the reaction, we adopted a catalytic reaction inspired by the oxidation of organic fuels in direct organic fuel cells. Pt-based catalysts used for these fuel cells lower the activation energy needed for the oxidation of organic fuels in acidic media. Similar to the fuel cell reactions, Pt catalysts can provide facile decomposition of ORA and electron transfer from ORA to $V^{4+}$. Consequently, by adjusting the amount of ORA, $V^{3.5+}$ electrolyte can be produced from $V^{4+}$ electrolyte by the catalytic reaction. As described in Fig. 1, the $V^{4+}$ electrolyte injected into a catalytic reactor incorporating Pt-based catalyst is simply converted to $V^{3.5+}$ electrolyte. This contrasts with the conventional electrolysis method in that electric energy is not consumed and surplus vanadium electrolyte is not generated. This emphasizes the merit of the catalytic production of $V^{3.5+}$ electrolyte. In this work, a concept for catalytic electrolyte production is proposed with rational material selections, and the feasibility of large-scale, continuous electrolyte production is proved with a prototype flow reactor. It was found the catalytic production of $V^{3.5+}$ electrolyte can reduce the production cost by 40% compared to conventional electrolysis method due to the process simplicity.

## Results

**Catalytic production of $V^{3.5+}$ electrolyte.** For use as a reducing agent for $V^{4+}$ solution, ORA should have a lower redox potential than that of $V^{4+}/V^{3+}$ (0.34 V vs. standard hydrogen electrode (SHE)), no remaining residue after the reduction, and low cost. Considering the requirements, methanol, formic acid, and oxalic acid are typically selected because their standard redox potentials are 0.02[25], −0.03, and −0.43 V, respectively, and their oxidation products are water and easily removable $CO_2$. As a proof of concept, the catalytic reduction of $V^{4+}$ was first demonstrated

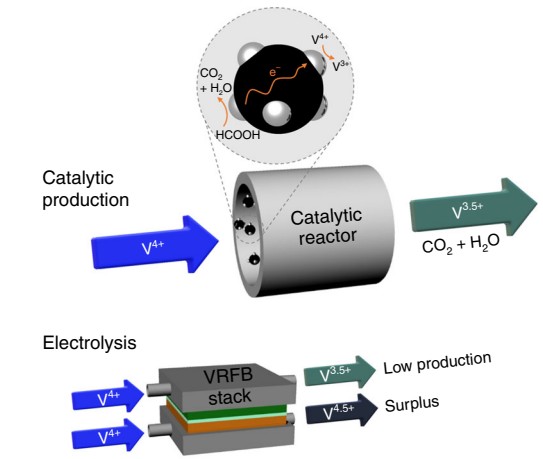

**Fig. 1** Schematics for producing the $V^{3.5+}$ electrolyte. The catalytic reaction (top) and conventional electrolysis (bottom)

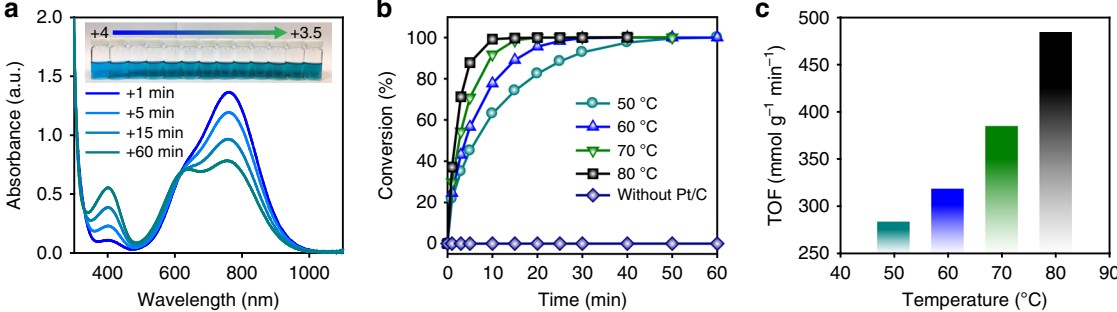

**Fig. 2** Catalytic reduction of $V^{4+}$ to $V^{3.5+}$. **a** Ultraviolet–visible (UV-Vis) spectra of the reactant electrolyte at different reaction times at 50 °C (inset: color changes of the reactant with the catalytic reaction). **b** Plots of the conversion with Pt/C at various reaction temperatures and that without Pt/C at 80 °C (100% conversion corresponds to the vanadium oxidation state of +3.5). **c** TOF (turnover frequency, mmol of $V^{3+}$ (g of Pt)$^{-1}$ min$^{-1}$) of the catalytic reaction at different reaction temperatures

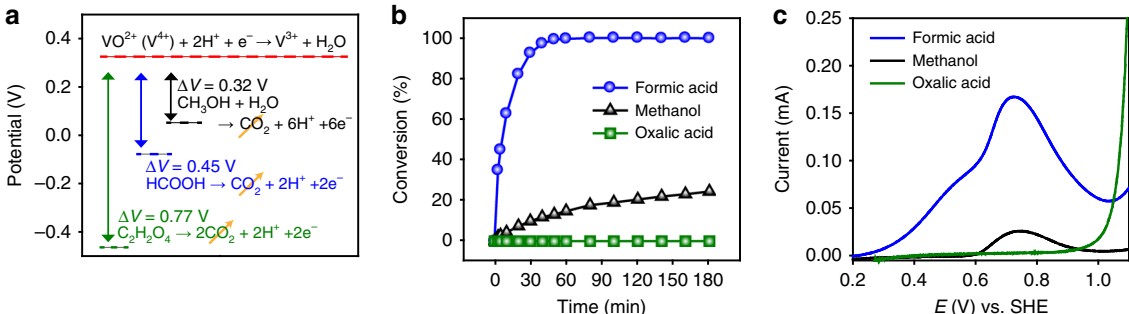

**Fig. 3** Effect of reducing agent on the catalytic reduction. **a** Standard redox potentials for the redox reactions of the reducing agents and that of the $V^{4+}/V^{3+}$ redox couple. **b** Conversion of the $V^{4+}$ electrolyte by formic acid, oxalic acid, and methanol. **c** Linear sweep voltammetry (LSV) results for oxidation of the reducing agents on the Pt/C-deposited rotating disk electrode

using formic acid and a typical fuel cell catalyst, Pt nanoparticles supported on carbon black (Pt/C).

After the addition of formic acid and Pt/C to the 1.55 M $V^{4+}$ electrolyte at 50 °C, the color of the electrolyte gradually changed from blue to dark green, indicating a decrease in the oxidation state of vanadium ions (inset of Fig. 2a)[6]. As shown in Fig. 2a, the ultraviolet–visible (UV–Vis) spectra of the electrolytes taken during the reaction exhibited a decrease in absorbance for $V^{4+}$ (760 nm) and an increase in absorbance for $V^{3+}$ (401 nm) with the reaction time, indicating conversion[26,27] from $V^{4+}$ to $V^{3+}$.

To verify the critical role of the Pt/C catalyst in the reaction, the conversion rates from $V^{4+}$ to $V^{3.5+}$ with and without Pt/C catalyst were determined using UV analysis and compared in Fig. 2b. It can clearly be seen that the $V^{4+}$ was readily reduced to $V^{3+}$ when Pt/C was incorporated, which contrasts with no sign of $V^{4+}$ reduction without Pt/C. This suggests that lowering the activation energy for formic acid oxidation is critical and that Pt/C is a requisite for chemical production of $V^{3.5+}$. Also displayed in Fig. 2b is that the reaction rate increased with the reaction temperature. The reaction rates at various reaction temperatures are expressed in terms of turnover frequency (TOF; mmol of $V^{3+}$ (g of Pt)$^{-1}$ min$^{-1}$; determined at 10% conversion, Fig. 2c). With the temperature increase from 50 to 80 °C, increase of TOF from 284 to 484 mmol of $V^{3+}$ (g of Pt)$^{-1}$ min$^{-1}$ was observed, suggesting that the production rate could be tuned by adjusting the reaction temperature.

Residue-free production was confirmed by checking the inclusion of any residual formic acid in the resulting electrolyte. According to the elemental analysis of the $V^{3.5+}$ electrolyte, the carbon content, which represents the amount of residual formic acid, was below a minimum detection level (<0.1% of total mass), confirming the high purity of the electrolyte prepared by the catalytic reaction.

**Effect of the ORA on the catalytic reduction of $V^{4+}$.** In order to find a rational guideline for selecting the best ORA, the catalytic reduction reactions for methanol, oxalic acid, and formic acid were compared. According to the redox potentials of the three ORAs, the thermodynamic driving force for the redox reaction increases in the order: methanol < formic acid < oxalic acid (Fig. 3a). However, the reaction activities of these ORAs are not consistent with their thermodynamic driving forces. Comparison of the $V^{4+}$ reduction reaction rates at 50 °C for the three ORAs with Pt/C catalyst (Fig. 3b) shows that formic acid has the highest reaction rate and oxalic acid the lowest. For formic acid, 100% conversion was achieved in an hour. However, for methanol, only 25% conversion had occurred after 3 h, and for oxalic acid, nearly no conversion was detected in spite of its largest thermodynamic driving force. This means that the catalytic reaction rate is not determined by the thermodynamic driving force, but by the activation energy needed for oxidation of the ORA.

The catalytic activity of Pt/C for the oxidation of these three ORAs was investigated using linear sweep voltammetry (LSV) with a Pt/C-deposited rotating disk electrode in $N_2$-purged 0.5 M $H_2SO_4$ (Fig. 3c). The LSV results show that the oxidation activity of ORA is strongly correlated with the $V^{4+}$ reduction rate. The onset potential for the oxidation of ORA was higher in the order: oxalic acid > methanol > formic acid; and the oxidation current was largest for formic acid, indicating that the highest catalytic

effect occurred with formic acid. This result matches well with previous studies about the catalytic oxidation of organic fuels on Pt catalyst[28–31]. The oxidation pathway for formic acid on Pt metal in a low potential region is known to include production of $CO_2$ without an intermediate CO generation step (called a "direct pathway"). This is in contrast to the CO generation and subsequent oxidation of CO by methanol. Therefore, formic acid oxidation is relatively easier than methanol oxidation, resulting in the lower onset potential. Pt/C catalyzes the oxidation of oxalic acid, to some extent, in comparison with a glassy carbon electrode (Supplementary Fig. 1). However, the onset potential was quite high due to the stable C–C bond of oxalic acid. According to previous research on oxalic acid oxidation on Pt, the difficulty in C–C bond cleavage explains the high onset potential of oxalic acid oxidation[32,33]. Based on these results, formic acid was selected as the ORA for the catalytic reduction of $V^{4+}$ electrolyte.

**Catalyst design for the catalytic reduction of $V^{4+}$.** There are many reports on catalysts to be used for oxidative decomposition of formic acid[34–39]. The heterogeneous Pd- and Pt-based catalysts are most commonly used because of their high activity. However, the use of Pd-based catalysts was rejected in this work because they show fast deactivation at the beginning of formic acid decomposition by a reaction intermediate[40–42] and weak stability under acidic conditions[43]. Therefore, we paid more attention to the Pt-based catalysts.

It is widely accepted that the poisoning of Pt catalyst by CO, which is a reaction intermediate of the electrochemical oxidation of methanol or formic acid, causes a decrease in reaction rates[44,45]. PtRu alloy catalyst is typically used to overcome the CO poisoning problem because Ru can easily dissociate water to produce metal oxide at lower potential than Pt, and thus help the oxidation of CO adsorbed on Pt by supplying an oxygen source[46–49]. This principle can also be applied to the catalytic reduction of $V^{4+}$, considering that the oxidation of formic acid determines the reaction rate. In this regard, we compared Pt/C and PtRu/C for $V^{4+}$ reduction at 50 °C with formic acid. As shown in Fig. 4a, PtRu/C exhibits much higher activity than Pt/C, given the use of the same amount of precious metals. This is in good agreement with previous results on the catalytic oxidation of formic acid. Chronoamperometry (CA) analysis confirmed the catalyst effect on formic acid oxidation. The oxidation currents in the initial period were lower for PtRu/C; however, after 3 min, they became larger than those of Pt/C (Fig. 4b). The initial high catalytic activity of Pt/C can be understood by considering that Pt has higher activity for formic acid decomposition than Ru does[50]. However, Pt/C gradually loses activity due to CO poisoning, meaning that PtRu/C provides higher activity than Pt/C as the reaction proceeds.

Stability of the catalyst is one of the factors important in designing the catalyst because VRFB performance is very sensitive to metal ion impurities in the electrolyte[13]. The chemical stabilities of Pt/C and PtRu/C in the $V^{4+}$ electrolyte were investigated by measuring the leached Pt and Ru ion concentration in the produced electrolyte using inductively coupled plasma mass spectrometer (ICP-MS) analysis after the catalytic reaction. As displayed in Fig. 4c, a relatively large amount of Ru ion was detected for the PtRu/C electrolyte, whereas the Pt ion concentrations of the Pt/C-derived electrolyte were within the noise level. These were confirmed by similar magnitudes of Pt and Ru signals for the electrolyte produced by electrolysis, which should not include any Pt and Ru residue. Therefore, PtRu/C possesses a stability issue under storage in acidic $V^{4+}$ electrolyte. To assess the stability of Pt/C during the catalytic reduction of $V^{4+}$, the catalytic reduction was repeatedly conducted without

change of the Pt/C catalyst and the change of catalytic activity was monitored in relation to the number of repetitions. As shown in Fig. 4d, the plot of conversion as a function of reaction time is nearly invariant regardless of the repetition number, indicating that the catalytic activity of Pt/C was preserved during repeated usage. After each repetition, the mean particle size and chemical composition of the Pt/C catalyst were investigated using transmission electron microscopy (TEM, Fig. 4e) and X-ray photoelectron spectroscopy (XPS, Fig. 4f), respectively. The raw data are provided in Supplementary Fig. 2. There was no significant change of Pt particle size or surface composition (Pt, $Pt^{2+}$, and $Pt^{4+}$) during repeated usage, which confirms the excellent stability of Pt/C under the redox reaction of $V^{4+}$ and in the acidic environment of formic acid.

The stability of Pt nanoparticle is quite reasonable considering the potential-pH diagram of Pt nanoparticle[51] (Supplementary Fig. 3). For 3-nm-sized Pt nanoparticle, the Pt dissolution potential below pH 0 is 0.860 V; therefore, Pt is stable in the form of metallic state if the equilibrium potential of Pt, which is determined by the redox potential of $V^{3+}/V^{4+}$ redox pair, is below 0.86 V. Since the standard redox potential of $V^{3+}/V^{4+}$ (0.340 V) is far lower than the Pt oxidation potential, the stability of Pt/C in the vanadium electrolytes is thermodynamically feasible.

**VRFB performances.** The quality of $V^{3.5+}$ electrolyte prepared by the catalytic reaction was assessed by investigating the electrochemical performance of a VRFB single cell at a current density of 80 mA $cm^{-2}$. The electrolytes prepared using the catalytic reaction with Pt/C or PtRu/C catalysts, and that prepared by electrolysis, were compared. Figure 5a shows the coulombic efficiencies (CEs) and energy efficiencies (EEs) of the cells upon cycling. The CEs and EEs for the electrolyte prepared with Pt/C (CE ~96%, EE ~88%) were nearly identical to those for the electrolyte from electrolysis, demonstrating that the quality of the electrolyte produced with Pt/C is equivalent to that of conventional electrolyte. However, the PtRu/C electrolyte showed abnormal fluctuations and low values for the efficiencies. It also exhibited heavy gas bubbling from the negative electrolyte, which was due to a hydrogen evolution reaction (HER) at the negative electrode during cell operation (Supplementary Fig. 4). The poor cell performance and hydrogen evolution for the electrolyte prepared with PtRu/C would be due to Ru ions leached from PtRu/C as Ru ions can be reduced to Ru metal on the carbon felt of the negative pole during charging and the Ru metal can work as a catalyst for the HER[52,53], resulting in the blockage of active sites and irreversible loss of capacity.

Comparison of the retention of discharge capacity with cycling (Fig. 5b) again demonstrated that the Pt/C-catalyzed electrolyte production is suitable to meet the required electrolyte quality and that the use of PtRu/C should be avoided. The electrolyte prepared with PtRu/C showed much smaller initial discharge capacity and rapid capacity fading due to the vigorous HER at the negative electrode. However, the Pt/C electrolyte and electrolysis electrolyte showed a nearly identical gradual decrease in capacity, which is mainly governed by vanadium ion crossover, and revealed equivalent qualities of the two electrolytes.

**$V^{3.5+}$ electrolyte production with a continuous flow reactor.** A prototype flow reactor employing a fixed bed of Pt/C catalyst was designed and constructed to demonstrate the feasibility of large-scale, continuous electrolyte production (Fig. 6a, b). The $V^{4+}$ electrolyte was continuously injected into a reactor held in a temperature-controlled water jacket, and the reduced electrolyte was ejected from the reactor to the external tank for storage. The

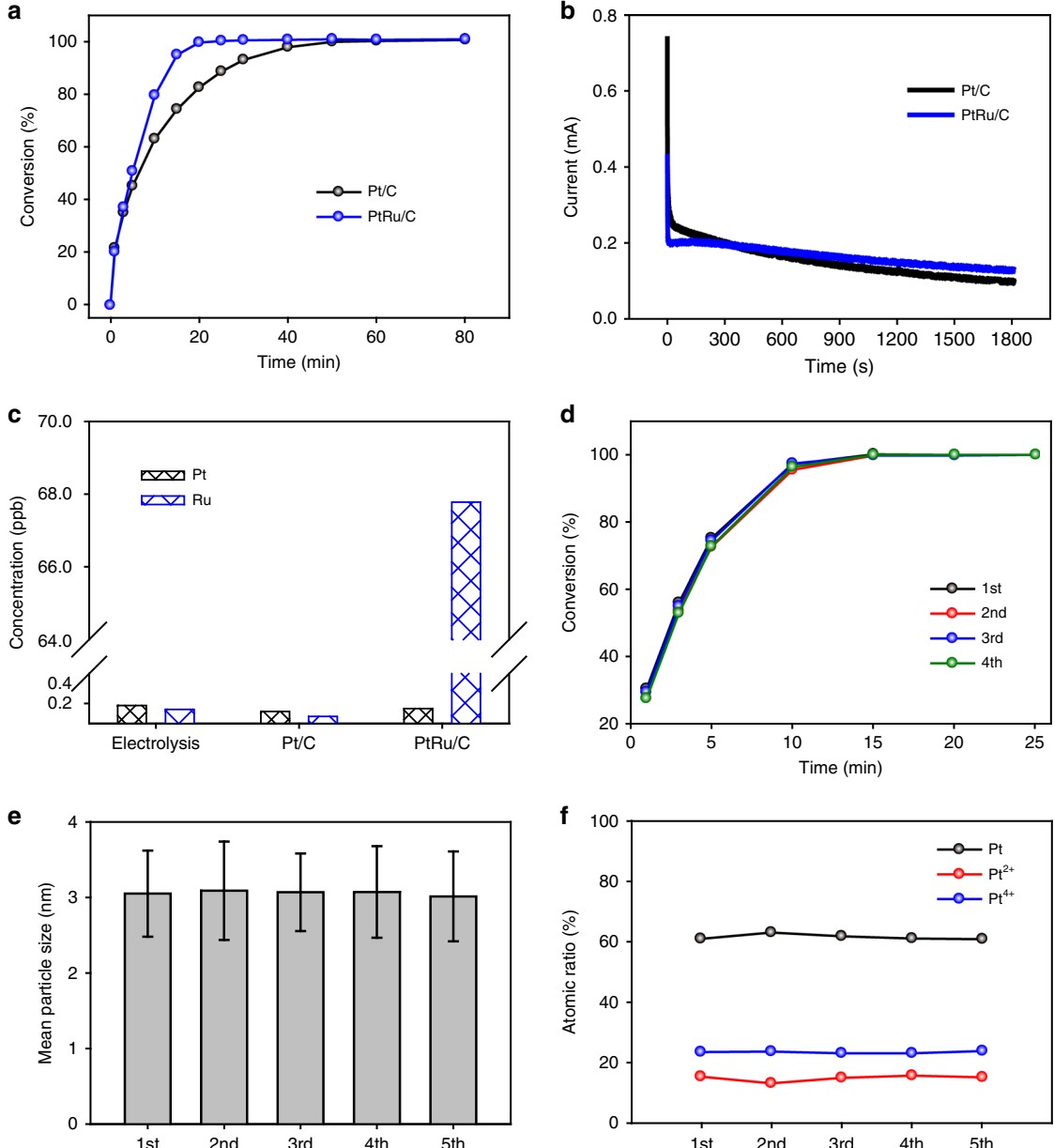

**Fig. 4** Effect of catalyst on the catalytic reduction. **a** Conversion of the $V^{4+}$ electrolyte with Pt/C and PtRu/C. **b** Chronoamperometry (CA) test for the oxidation of formic acid on Pt/C and PtRu/C electrodes; stability of Pt/C and PtRu/C for the catalytic reduction. **c** Inductively coupled plasma (ICP) analysis results for the diluted electrolytes from electrolysis and catalytic reaction with Pt/C and PtRu/C at 80 °C for 30 min. **d** Plots of conversion as a function of reaction time for the repeated catalytic reductions at 80 °C without refreshing Pt/C. Variation of **e** mean particle size and **f** surface composition of Pt/C with the repeated usage, as determined by transmission electron microscopy (TEM) and X-ray photoelectron spectroscopy (XPS) analysis, respectively. The error bars represent the standard errors. Source data are provided as a Source Data file

reactor consisted of a cylindrical polytetrafluoroethylene (PTFE) container with 575 cm$^3$ of inner volume and a stack of 30 carbon-felt sheets (each 17.7 cm$^3$) that accommodated 3.1 g of Pt/C. Carbon felt, which is conventionally used as an electrode for VRFB, was chosen as the substrate upon which to fix Pt/C particles because of its large surface area, small flow resistance, high chemical stability in VRFB electrolyte, and high electron conductivity. Due to its high electron conductivity, carbon felt allows rapid electron transfer from formic acid to $V^{4+}$, facilitating the reduction of $V^{4+}$. To immobilize Pt/C particles on the carbon felt, a slurry of Pt/C and Nafion was coated and dried on the carbon-felt surfaces. The formation of a uniform Pt/C layer on the carbon felt was confirmed by scanning electron microscopy (SEM), as shown in Fig. 6c, d. The high macro-porosity of the carbon felt

was preserved after the surface decoration, ensuring efficient flow of the electrolyte through the carbon-felt stack. The reactor was designed as an air-tight system to prevent air intrusion because oxygen, in the presence of Pt catalyst, can accelerate a reverse reaction (i.e., oxidation of $V^{3+}$ to $V^{4+}$, Supplementary Fig. 5).

The reactor was successfully operated as displayed in Supplementary Movie 1. During operation with production rate of 2 L h$^{-1}$, the product electrolytes were sampled and analyzed using UV–Vis spectroscopy. The oxidation state of the electrolyte was about +3.5 and invariant during the continuous production, as shown in Fig. 6e. Even after 30 times operations, the oxidation state of the product electrolyte was not changed (Fig. 6f). This indicated the stability of the Pt/C-decorated carbon felt and other components of the reactor. The $V^{3.5+}$ electrolyte produced by the

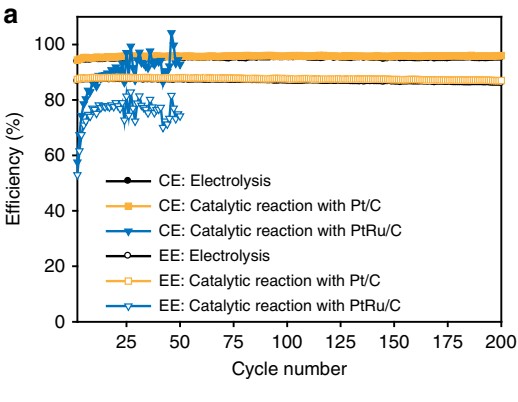

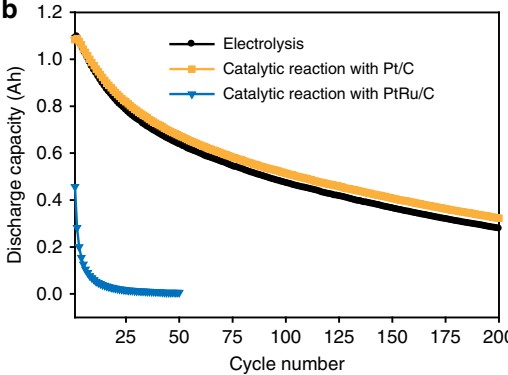

**Fig. 5** Electrochemical performance of the $V^{3.5+}$ electrolyte. **a** Coulombic efficiencies (CEs), energy efficiencies (EEs), and **b** discharge capacities of cells cycling at 80 mA cm$^{-2}$ using electrolytes produced by catalytic reaction using Pt/C and PtRu/C, and by electrolysis

reactor exhibited stable CEs (95%) and EEs (86%) during 200 cycles in the range of 0.80–1.65 V (Fig. 6g), which is equivalent to the results with electrolyte produced by the small-scale batch reaction (Fig. 5).

As mentioned earlier, the stability of Pt catalyst during the catalytic production is highly important to achieve high VRFB performances because the dissolved Pt ion can cause hydrogen evolution at the negative electrode, lowering efficiency and cycling stability. Since the reduction of Pt$^{2+}$ to Pt and subsequent hydrogen evolution are accelerated with increasing states of charge (SOC), the cells were operated in a wider voltage window of 0.7–1.9 V to further confirm the absence of the Pt dissolution. The charge cut-off voltage of 1.9 V leads to a higher SOC of 99% (Supplementary Note 1)[7,54]. As it is shown in Fig. 7a, the reactor electrolyte showed identical efficiencies with the electrolysis electrolyte without any discrepancies even at the high SOC condition, proving the absence of any side reactions. Since HER can take place even in the rest stage when solution is in high SOC, self-discharge test was also conducted to garner consolidated conclusion about the high stability of Pt catalyst in the reactor. The reactor electrolyte manifested negligible differences with the electrolysis electrolyte in the self-discharge test after the charge up to 2.4 V (Fig. 7b), again verifying that Pt ion is not included in the reactor electrolyte by the rationale from excluded HER.

## Discussion

A method of preparing $V^{3.5+}$ electrolyte for VRFB using catalytic reaction of ORA was demonstrated in this report. Candidates for ORA were selected according to logical guidelines, and among the candidates, formic acid was chosen as the best ORA. This is because formic acid showed a fast reaction rate without leaving any residues, while methanol and oxalic acid showed sluggish

reaction. The catalyst was selected by comparing the activities and stabilities of Pt/C and PtRu/C. Even though PtRu/C showed higher activity than Pt/C, it was rejected due to Ru dissolution that resulted in extreme hydrogen gas generation during cell operation. However, Pt/C showed superior stability and electrolyte produced using Pt/C showed stable cell operation (CE ~96%, EE ~88%), indicating that Pt/C is an appropriate catalyst for the catalytic production of $V^{3.5+}$ electrolyte. A prototype continuous flow reactor was designed to validate the possibility of large-scale production by the catalytic reaction. Pt/C-decorated carbon-felt sheets were stacked to build a fixed-bed catalyst reactor. The reactor showed stable operation over 30 h, and the produced electrolyte showed excellent cell performance. Therefore, we successfully provided evidence for the feasibility of large-scale catalytic production of $V^{3.5+}$ electrolyte.

The catalytic process is highly cost-effective in terms of the process simplicity. The detailed cost analysis was conducted for the catalytic production and industrial electrolysis. For a comparison of capital cost with large-scale electrolysis using VRFB stack, the cost of a catalyst reactor with 40 L h$^{-1}$ production capacity is provided in Supplementary Note 2. Since the vanadium materials used for the two processes are identical, the production cost comparison evaluated the reducing agent, feedstock, energy, and consumable materials costs. It was found that the proposed catalytic process could reduce the production cost by 40% compared to the industrial electrolysis process (Supplementary Note 2) due to the simplicity of the process. It is believed that future optimization of the reactor technology can promise the further reduction of the production cost of VRFB electrolyte.

## Methods

**Catalytic reduction of $V^{4+}$ electrolyte**. To proceed with catalytic reduction of $V^{4+}$ electrolyte, a $V^{4+}$ electrolyte (1.55 M $V^{4+}$ in 4.17 M SO$_4^{2-}$) was heated to the desired temperature; then, catalyst and ORA were successively added to the electrolyte with vigorous stirring. The molar ratio of ORA to $V^{4+}$ was 0.27 for formic acid (95%, Sigma-Aldrich), 0.09 for methanol (99.6%, OCI Company, Ltd.), and 0.13 for oxalic acid (99%, Sigma-Aldrich), which corresponds to about 8% excess of the theoretically required amount for producing $V^{3.5+}$. To avoid the oxidation of $V^{3+}$ to $V^{4+}$, coupled with the reduction of oxygen in the presence of Pt-based catalysts, the reaction was conducted in nitrogen gas atmosphere. The catalysts used in experiments included Pt/C (46.4 wt% Pt, Tanaka Kikinzoku Kogyo) and PtRu/C (29.8 wt% Pt, 23.1 wt% Ru, Tanaka Kikinzoku Kogyo). The slight excess of ORA was used to compensate for the reverse reactions due to residual oxygen in the electrolyte. The amounts of Pt and PtRu were 0.382 g with respect to 1 mol of $V^{4+}$. During the reaction, the reaction solution was sampled regularly and the vanadium oxidation state of the sample solutions was determined using a UV–Vis spectroscopy (GENESYS$^{TM}$ 10s) after dilution. The resulting $V^{3.5+}$ electrolyte was filtered through a hydrophilic PTFE membrane (Omnipore$^{TM}$; 0.2 μm pore size) to remove the catalyst particles from the electrolyte; this procedure was repeated if needed.

**Electrochemical analysis of the catalytic oxidation of ORAs**. A three-electrode electrochemical cell, including a catalyst-coated glassy carbon (working electrode), a platinum wire (counter electrode), and an Ag/AgCl (3 M KCl) reference electrode were used to characterize the catalytic oxidation of ORA. Electrode potentials, for simplicity, were expressed with respect to a SHE. Catalyst-coated glassy carbon electrodes were prepared by disposing 10 μL of finely dispersed Pt/C catalyst ink (0.17 wt% Pt/C, 0.02 wt% Nafion, 0.19 wt% 1-propanol, 74.55 wt% water, and 25.07 wt% isopropyl alcohol) or a PtRu/C catalyst ink (0.15 wt% PtRu/C, 0.02 wt% Nafion, 0.16 wt% 1-propanol, 74.6 wt% water, and 25.07 wt% isopropyl alcohol) on a glass disk electrode. This was followed by drying at ambient temperature for 30 min with rotation at 700 r.p.m. By using the catalyst-coated electrodes, it was possible to conduct LSV and CA tests in N$_2$-purged 0.5 M H$_2$SO$_4$ electrolyte with various reducing agents (0.42 M of formic acid, 0.42 M of oxalic acid, and 0.14 M of methanol). The LSV test was measured within the potential range of 0.05–1.2 V vs. SHE at a scan rate of 0.1 mV s$^{-1}$. CA analysis was conducted at 0.45 V for formic acid. During the LSV and CA tests, the catalyst-coated electrodes were rotated at 1500 r.p.m. to remove the produced CO$_2$ from their surfaces.

**Characterization of the electrolytes and catalysts**. The amounts of Pt and Ru ions in the prepared electrolytes (1.55 M of $V^{3.5+}$ in 4.17 M of SO$_4^{2-}$), which can be leached during the catalytic reduction, were quantified using an ICP-MS (ELAN DRC II). The presence of residual ORAs in the electrolytes was checked using an

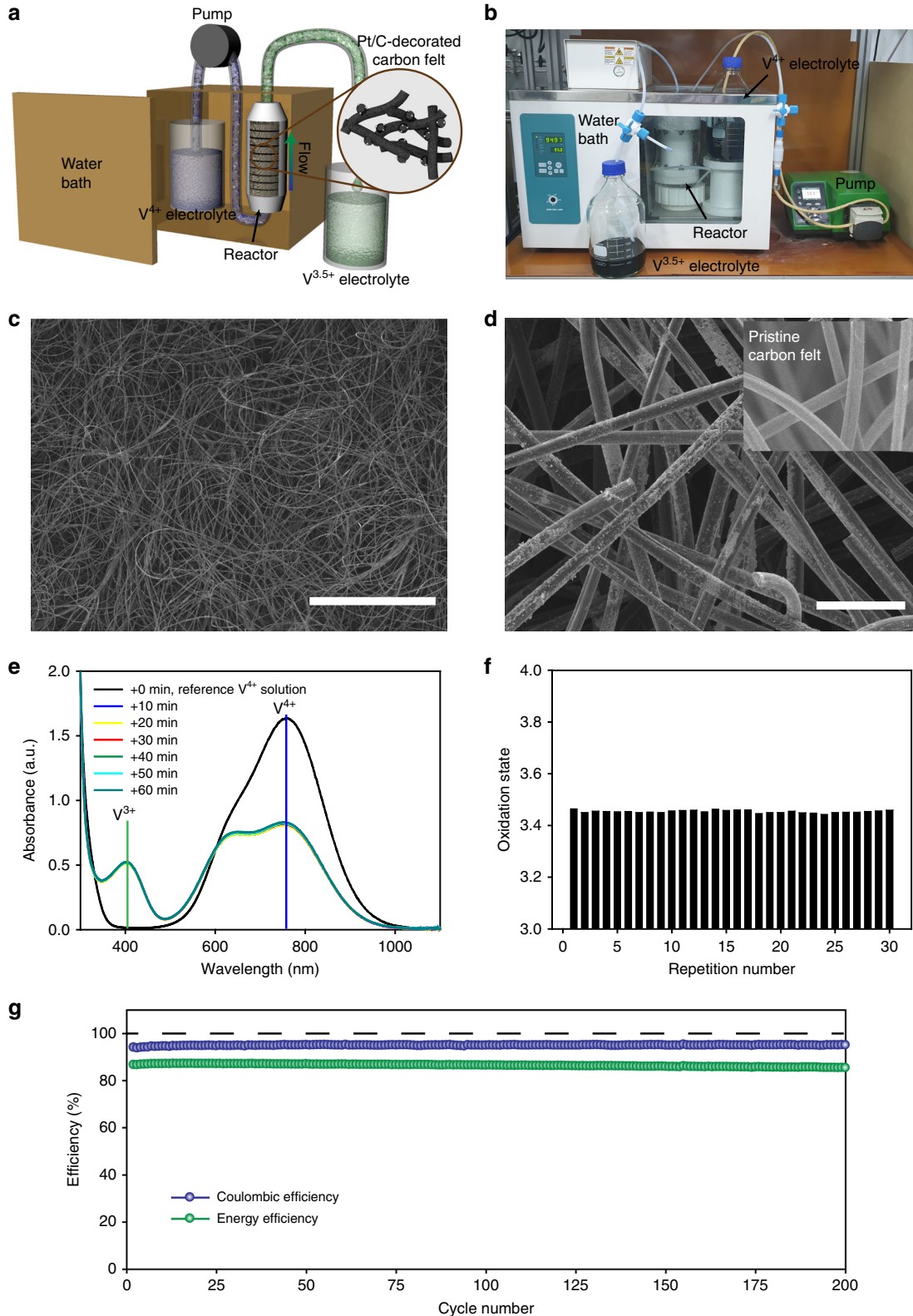

**Fig. 6** Continuous production of the $V^{3.5+}$ electrolyte by the catalytic reactor. **a** Structure and **b** optical image of the catalytic reactor. Scanning electron microscopy (SEM) image of the Pt/C-decorated carbon felt at **c** low and **d** high magnification. (Scale bars: **c** 1 mm, **d** 50 μm. Inset: SEM image of the pristine carbon felt in the same scale with (**d**)). **e** Ultraviolet–visible (UV–Vis) spectra of the feed ($V^{4+}$ electrolyte) and product electrolytes taken at different operation times during continuous 2 L h$^{-1}$ catalytic reaction. **f** Oxidation states of the product electrolytes from the 30 repeated operations. **g** Coulombic and energy efficiencies of the cell with its electrolyte produced using the catalytic reactor

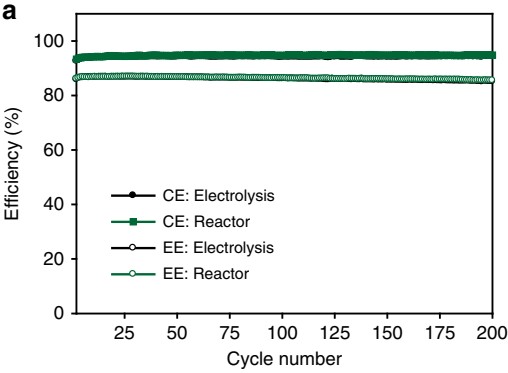

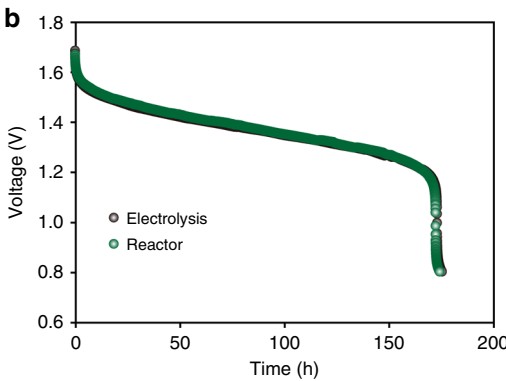

**Fig. 7** Electrochemical stability of the reactor electrolyte. **a** Cell performances of the electrolytes prepared by the catalyst reactor and electrolysis under a wider voltage window (0.7–1.9 V). **b** Self-discharge test for the reactor and electrolysis electrolytes after the charge up to 2.4 V

elemental analyzer (EA, FLASH 2000 series). For the ICP-MS analysis, the electrolytes were diluted by a factor of 100 with deionized water. For the EA analysis, the electrolytes were adjusted to pH 3 using NaOH.

After the catalytic reduction reaction, the chemical and physical structures of Pt/C were investigated using TEM (Tecnai F20 and Tecnai G2 F30 S-Twin, FEI Company) and XPS (Al K-alpha, Thermo VG Scientific).

**VRFB single-cell test.** A single cell having a 6 cm$^2$ active electrode area was used to monitor cell cycling performance. The cell consisted of a Nafion 115 membrane, two pre-heat-treated carbon-felt sheets (GFD4.6, SGL Carbon Group), two graphite mono-polar plates, two copper current collectors, and two end plates. A control V$^{3.5+}$ electrolyte was prepared by the conventional electrolysis method from V$^{4+}$ electrolyte[55]. Cell tests were conducted with 30 mL of the vanadium electrolyte (1.55 M of V$^{3.5+}$ in 4.17 M of SO$_4^{2-}$) for both positive and negative compartments. Charging/discharging tests were performed at 80 mA cm$^{-2}$ and at a flow rate of 30 mL min$^{-1}$ in the voltage range 0.8 to 1.65 V using a test station (WBCS3000, WonATech).

**Fabrication of the flow reactor.** A prototype flow reactor was prepared by stacking Pt/C-decorated carbon-felt sheets in a PTFE cylinder. The Pt/C-decorated carbon felt was prepared by dipping a 4.6-mm-thick heat-treated (520 °C in air for 9 h) carbon felt (GFD4.6, SGL Carbon Group) in catalyst ink consisting of 0.65 wt % Pt/C, 0.35 wt% Nafion, 49.49 wt% water, and 49.51 wt% n-propyl alcohol. This was dried at room temperature for 12 h, and then further dried at 60 °C under vacuum for 12 h. The areal Pt loading level of the Pt/C-decorated carbon felt was controlled to be 2.65 mg cm$^{-2}$. A rubber gasket was inserted between the Pt/C-decorated carbon-felt sheets to prevent leakage of the electrolyte. By using a peristaltic pump (530S, Watson-Marlow), the V$^{4+}$ electrolyte was injected into the reactor (at 95 °C) at a volumetric flow rate of 1–2 L h$^{-1}$. A SEM (SU8230, HITACHI) was used to image the Pt/C coating on the carbon fibers of the carbon felt.

## Data availability

The data that support the findings of this study are provided in the published article and/or its Supplementary Information files. The source data underlying Fig. 4e are provided as a Source Data file, and all datasets are available from the corresponding author upon reasonable request.

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

## Acknowledgements

This work was supported by the Korea Institute of Energy Technology Evaluation and Planning (KETEP) and the Ministry of Trade, Industry, and Energy (MOTIE) of the Republic of Korea (No. 20172420108480).

## Author contributions

J.H. and H.-T.K. conceived the concept of catalytic production of impurity-free $V^{3.5+}$ electrolyte for VRFB, and designed this work. J.H. carried out the main experiments, including experimental planning, electrochemical measurements, and data analysis; J.H. and J.-Y.H. prepared flow reactor; J.-Y.H. and S.-K.R. conducted continuous production of $V^{3.5+}$ electrolyte test using flow reactor and modified reactor performance; S.K. and S. Y. participated in experimental planning and provided experimental insights; C.C., R.K. and J.-H.L. assisted with sample preparation and discussion of the results; A.K. worked on the cost analysis of the proposed method; J.H. and H.-T.K. wrote the manuscript, and H.-T.K. and S.-K.R. supervised this work; all authors commented on the manuscript.

## Competing interests

The authors declare no competing interests.
