## [Peer Review File · Nature Communications]

Reviewers' comments:

Reviewer #1 (Remarks to the Author):

This paper describes an improved method to produce electrolyte for the vanadium redox battery using a Pt-based catalysts for the chemical reduction of V(4) solutions. While some scientifically interesting results are presented, there is insufficient information provided to fully assess the commercial viability of this process. The authors also fail to acknowledge the original work performed and published by Skyllas-Kazacos and co-workers on a range of chemical and electrolytic methods to produce vanadium electrolyte from various insoluble vanadium compounds in the 1980s as detailed in the following patent.

"Vanadium salt dissolution process", M. Skyllas-Kazacos and R. McDermott, Patent Appl. No. PCT/AU88/00471, Sth. African Patent No. 88/9244

Although the above patents was mentioned in relation to its use of residue free organic reducing agents, there is no discussion of the other electrolytic methods proposed here and in the subsequent patent:

"High Energy Density Vanadium Electrolyte Solutions, Methods of Preparation Thereof and All-Vanadium Redox Cells and Batteries Containing High Energy Density Vanadium Electrolyte Solutions", M. Skyllas-Kazacos, International Patent Application No. PCT/AU96/00268, May, 1996, United States Patent 7078123

Using the methods described in the above patent, a pure V(3.5+) solution is easily produced without excess V(5+) as suggested by the authors in the text and in Figure 1. Their suggestion that the currently available methods are inefficient and uneconomic is therefore very misleading. Their proposed catalytic method needs to be put in proper context and compared in technical and economic terms with the methods described in the above earlier patents;

The method described in this paper involves the use of a Pt-based catalyst to chemically reduce V(4+) with formic acid. While Pt is shown to increase the kinetics of this reaction, a very important consideration has been overlooked. The authors claim that an impurity free electrolyte can be produced by this method, but the possibility of leaching Pt into the electrolyte during extended use of the catalyst has not been fully investigated over prolonged use under industrial conditions. As acknowledged by the authors, Pt and other noble metals are known to catalyse the V(2+) + H+ reaction that displaces protons from solution, producing hydrogen gas. This reaction can occur even when the fully or partially charged battery is in standby mode and is a very undesirable reaction that needs to be avoided.

For the ICP-MS analysis, the electrolytes were diluted by a factor of 100 with DI-water. This dilution could have led to any leached Pt remain undetected.

The charge-discharge tests were performed between the voltage range 0.8 to 1.65 V but depending on the cell resistance, this may represent a low state-of charge at the end of the charge cycle. The impact of excess hydrogen evolution may not therefore be seen at lower SOCs. The authors need to provide typical voltage vs time charge discharge curves and give the open-circuit potentials at the end of the charge cycle to demonstrate that high SOCs are being reached during these cycling tests. The solution potential vs time should also be monitored for several days for a fully charged solution to verify that no hydrogen ion displacement reaction is occurring that leads to hydrogen evolution on standing at high SOCs.

Before this paper can be accepted for publication in Nature Communications therefore, more information should be provided as detailed above.

Reviewer #2 (Remarks to the Author):

The cost of producing the vanadium-based electrolyte for redox flow batteries is one of the main causes of the still higher storage costs compared to lithium ion batteries, for example. The proposed

method could certainly contribute to the required cost reduction. However, in order to be able to evaluate this, at least a cost estimate should be made in comparison to the electrolysis method, taking into account the much more complex reactor and the high costs for the catalyst used.

Minor changes:

Line 176: The heterogeneous (remove: catalysts) Pd- and Pt-based catalysts are most commonly used because 176 of their high activity.

Reviewer #3 (Remarks to the Author):

The authors propose a simpler method for chemical production of impurity-free V^{3.5+} electrolyte by utilizing organic fuel cell catalysts. The new method features the use of catalyst and an organic reducing agent (ORA), which enable the reduction of V⁴⁺ to V³⁺ without releasing any undesired impurities. However, the major bottleneck for all vanadium redox flow battery is the materials cost of vanadium itself, in addition, this method uses Pt/C in the process, which is very expensive (precious metals). I do not see economical advantage of using this method. The authors should provide more in-depth analysis to justify that this method will address the high cost of Vanadium electrolyte. For example, how much cost reduction can be achieved by using the proposed method? Is precious metal required in this process? If so, what's the implication for price? From my understanding, the vanadium redox flow battery using conventional method can perform as good as what is reported in the manuscript if not better. Therefore, I do not find strong justifications to support the publication of this manuscript at its present form.

Response Letter

Title: Catalytic production of impurity-free $V^{3.5+}$ electrolyte for vanadium redox flow batteries

We highly appreciate the reviewers' comments on our manuscript entitled "Catalytic production of impurity-free $V^{3.5+}$ electrolyte for vanadium redox flow batteries". Every comments were highly meaningful and certainly helped us to improve the quality of our manuscript. Reflecting the reviewers' comments and advises, we revised the manuscript and supplementary information. We hope this revision fully address the issues raised by the reviewers. Changes made in the manuscript and supplementary information are indicated with **yellow background**.

Reviewer #1 (Remarks to the Author):

This paper describes an improved method to produce electrolyte for the vanadium redox battery using a Pt-based catalysts for the chemical reduction of V(4) solutions. While some scientifically interested results are presented, there is insufficient information provided to fully assess the commercial viability of this process. The authors also fail to acknowledge the original work performed and published by Skyllas-Kazacos and co-workers on a range of chemical and electrolytic methods to produce vanadium electrolyte from various insoluble vanadium compounds in the 1980s as detailed in the following patent.

"Vanadium salt dissolution process", M. Skyllas-Kazacos and R. McDermott, Patent Appl. No. PCT/AU88/00471, Sth. African Patent No. 88/9244

Although the above patents was mentioned in relation to its use of residue free organic reducing agents, there is no discussion of the other electrolytic methods proposed here and in the subsequent patent:

"High Energy Density Vanadium Electrolyte Solutions, Methods of Preparation Thereof and All-Vanadium Redox Cells and Batteries Containing High Energy Density Vanadium Electrolyte Solutions", M. Skyllas-Kazacos, International Patent Application No. PCT/AU96/00268, May, 1996, United States Patent 7078123

Using the methods described in the above patent, a pure $V(3.5+)$ solution is easily produced without excess $V(5+)$ as suggested by the authors in the text and in Figure 1. Their suggestion that the currently available methods are inefficient and uneconomic is therefore very misleading. Their proposed catalytic method needs to be put in proper context and compared in technical and economic terms with the methods described in the above earlier patents;

Our response: We sincerely appreciate for the review's comments on our manuscript.

As kindly suggested, the patent "High energy density vanadium electrolyte solutions, methods of preparation thereof and all-vanadium redox cells and batteries containing high energy density vanadium electrolyte solutions" was added to the references of the revised manuscript in relation to the use of residue free organic reducing agents. Also, more pioneering works of Prof. Skyllas-Kazacos were added to the references [7,8,9] to help the readers recognize her valuable contributions to all-vanadium redox flow battery technology.

The reviewer pointed out that an electrolysis method using water splitting reaction (oxygen evolution reaction (OER)) as counter reaction for the V^{4+} reduction (Patent Application US 10/226,751 (2006)) was omitted in the discussion of current method for producing $V^{3.5+}$ electrolyte without surplus V^{5+} electrolyte. This method utilizes an electrolytic cell which can reduce V^{5+} to $V^{3.5+}$ at negative electrode and oxidize water to form oxygen at positive electrode. However, the current membrane technology cannot perfectly prevent the crossover of vanadium ions from the negative electrode to positive electrode in the electrolysis cell, resulting in an inevitable loss of vanadium species. The vanadium ions crossing over the membrane from the negative to positive electrode are subject to oxidation. In fact, the patent states the vanadium oxidation reactions to generate V^{4+} and V^{5+} happen at the positive electrode also as well as OER (section 71, Patent Application US 10/226,751 (2006)) as described below.

At positive electrode **12**, the following reactions take place:

Therefore, this method either wastes vanadium materials or lowers electrolysis efficiency in the production of $V^{3.5+}$ electrolyte.

Additionally, the OER requires high oxidation potential, which can incur rampant carbon corrosion diminishing the lifespan of the cell or stack (*Angew. Chem. Int. Edit.* **53**, 10960 – 10964 (2014)). In the technology sector of electrolysis-based hydrogen production, the carbon corrosion of OER electrode under acidic condition still remains challenge. Therefore, expensive titanium foil and mesh are used as a corrosion-resistant current collector and diffusion electrode, respectively. Expensive RuO_2/IrO_2 catalyst are used for OER to get high current density operation. The use of expensive membrane also imposes a burden on cost reduction of electrolysis.

The patent includes a membrane-free cell. In this case, the vanadium oxidation at the OER electrode becomes more significant, and the oxygen formed at the positive electrolyte again oxidize V^{3+} , hindering the conversion from V^{4+} to V^{3+} .

Fig. R1 $V^{3.5+}$ electrolyte production test by using water electrolysis: (a) Voltage profile of the electrolysis cell coupled with OER and V^{4+} reduction operated at a current density of 50 mA cm^{-2} . (b) UV-Vis spectra of the anolytes (OER side) taken during the electrolysis. (c) Optical image of the bipolar plate of the anode (left) and the cathode (right) after the electrolysis for 3.5 h. SEM image of carbon felt at (d) anode and (e) cathode after the $V^{3.5+}$ production using water splitting reaction. Inset image in (e) shows SEM image of pristine carbon felt.

Fig. R1 shows the result of $V^{3.5+}$ electrolyte preparation using the water electrolysis introduced in the patent that the reviewer commented to refer. The experiments were conducted with the same V^{4+} electrolyte used in this work (catholyte) and sulfuric acid solution (anolyte). All the above-mentioned problems related with OER were clearly observed in our test. The production of $V^{3.5+}$ electrolyte using water electrolysis suffered from the high operation voltage (**Fig. R1a**), crossover of vanadium ions (**Fig. R1b**), and corrosion of carbon materials (bipolar plate and carbon felt electrode, **Fig. R1c ~ e**). The severe damages of the cell components at OER side due to carbon corrosion were salient (**Fig. R1c (left side)**, **Fig. R1d**), and it is believed the use of expensive Ti based cell components are imperative. Also, a continuous loss of vanadium ion was identified from the increase of UV-Vis absorbance below 300 nm for the anolyte, which curtails the benefits of the electrolysis method for the preparation of $V^{3.5+}$ electrolyte.

We expect these problems might be addressed by advanced engineering. However, we would like to emphasize that the catalytic reactor-based electrolyte production, intrinsically, does not have such disadvantages of 1) loss of vanadium material at OER electrode (generation of V^{5+}), 2) use of Ti foil and/or mesh for OER electrode, 3) use of expensive $\text{RuO}_2/\text{IrO}_2$ catalyst, and 4) use of expensive membrane.

Table R1 Comparison of the processes for preparing $V^{3.5+}$ electrolyte

Technology	Catalytic reduction (this work)	Electrolysis (Positive : V^{4+}/V^{5+}) (Negative : V^{4+}/V^{3+})	Electrolysis (Positive : OER) (Negative : $V^{5+}/V^{4+}/V^{3+}$)
Reactor	Catalytic reactor	Stack	Stack
Electrode	Carbon felt	Carbon felt	Positive : Ti-based electrode Negative : Carbon felt
Catalyst	Pt/C	-	RuO_2 , IrO_2 for OER
Membrane	Not used	Ion exchange membrane	Ion exchange membrane
Surplus V^{5+}	Not formed	Formed	Formed. Its amount depends on the degree of crossover through membrane.

The catalytic reduction and two electrolysis methods are compared in the **Table R1**. As indicated, we believe our new VRFB electrolyte production method has key advantage of not generating surplus V^{5+} , not wasting vanadium materials, and not using expensive current collector and membrane.

Unfortunately, the cost comparison between the catalytic reaction and the electrolysis method using water splitting was unable for us due to only limited information about the electrolysis method with water splitting reaction in the patent. However, we could compare the electrolyte production costs for the newly proposed catalytic reduction method and the electrolysis method currently employed by Avalon Battery Inc.

**Fig. R2** Process diagrams for the production of 1.55 M $V^{3.5+}$ electrolyte by (a) catalytic reaction, and (b) conventional electrolysis method.

The process diagram of the newly proposed process for $V^{3.5+}$ electrolyte preparation using the catalytic reaction is given in **Fig. R2a**. Starting from V^{4+} electrolyte, the $V^{3.5+}$ electrolyte production can be achieved by a single step reaction using the catalytic reactor in a continuous

manner. The conventional process modelled used the flow chart of **Fig. R2b**. It uses chemical reduction of V^{5+} to V^{4+} with SO_2 , followed by electrolytic reduction to $V^{3.5+}$ finished product. Note that in this conventional process, it is possible to avoid the higher capital costs of electrolyzers using the OER counter electrode described in the reference (Patent Application US 10/226,751 (2006)). Instead, a conventional VRFB cell stack with carbon felt electrodes can be used for both anolyte and catholyte. The counter-electrode is fed with the same V^{4+} material which is converted to $V^{4.5+}$ at the counter electrode. The $V^{4.5+}$ stream is then recycled to the front of process, reduced to V^{4+} and reprocessed. This recycle stream does result in the need for larger equipment and some additional processing cost.

Table R2 Cost of the catalyst reactor (Production capacity: 40 L h⁻¹)

	Material	Specification	Unit	Unit cost (\$)	Usage	Cost (\$)
Reactor	Teflon body	ϕ 300 * ϕ 220 * 300L	ea	625	2	1,250.0
	Teflon cap	ϕ 300*105	ea	419.2	2	838.3
	1/4" male connector		ea	2.9	2	5.8
	SUS bolt	M8*55	ea		30	16.7
	Cost of labor					166.7
Sub total						2,277.5
Pt/C-decorated carbon felt	Pt/C	TEC10E50 (Tanaka)	g	20.5 ^{a)}	62	1,271.0
	Ionomer	D520 (DuPont)	L	316.7	0.72	228.0
	N-propyl alcohol		L	3.2	5.5	17.4
	Water		L	0.0033	4.5	0.015
	Carbon felt	GFD4.6 (SGL)	m ²	91.7	2.35	215.4
Sub total						1,731.8
Total						4,009.3

a) Tanaka TEC10E50: 2460YEN/g

For a fair comparison with the large-scale electrolysis production using VRFB stack, catalyst reactor with production capacity of 40 L h⁻¹ was derived by the experimental results we obtained from the lab-scale prototype reactor. The large-scale reactor cost was analyzed as shown in **Table R2**. The reactor mainly consists of a Teflon-based housing and Pt-decorated carbon felts. The catalytic activity of the reactor was measured to be 0.645 L g_{Pt/C}⁻¹ h⁻¹, therefore, the required Pt/C catalyst for the reactor was 62 g (Pt: 28.8 g). Considering the amount of the catalyst and other components, the total cost of the reactor, which is capable of the production at 40 L h⁻¹, is \$ 4,009 (US dollar). One can compare the stack cost at the same production capacity of 40 L h⁻¹ to identify the capital cost difference.

Table R3 Process cost for the catalytic production of 1 L of 1.55 M V^{3.5+} electrolyte

Component	Requirement	Unit	Unit cost	Cost (\$ L ⁻¹)
Formic acid	0.3875	mol L ^{-1 c)}	0.088 \$ mol ⁻¹	0.0341
Electrical heat input ^{a)}	55.14	Wh L ⁻¹	0.15 \$ kWh ⁻¹	0.0083
Consumable ^{b)}	-	-	-	0.0072

Total	0.0496
a) Assumptions	
1. Heat capacity of electrolyte: 3.7 kJ kg ⁻¹ °C ⁻¹	
2. Heat recovery: 0.5 (fraction of total heat needed to raise temperature from 21°C to 95°C for reaction)	
3. Electricity cost: 0.15\$ kWh ⁻¹ (The same as that for the electrolysis)	
b) Assumptions: Pt/C coated electrode need to be replaced every 6000 h operation.	
c) L ⁻¹ in every unit refers to liter of vanadium solution.	

Table R4 Process cost for the conventional electrolysis production of 1 L of 1.55 M V^{3.5+} electrolyte

Component	Requirement	Unit	Unit cost	Cost (\$ L ⁻¹)
Reducing agent ^{a)}	0.04	kg L ^{-1 c)}	1.2 \$ kg ⁻¹	0.0480
Electricity consumption	0.17	kWh L ⁻¹	0.15 \$ kWh ⁻¹	0.0255
Consumable ^{b)}	-	-	-	0.0080
Total				0.0815
a) SO ₂ was used as reducing agent.				
b) Assumptions: Stack electrode need to be replaced every 500,000 L VRFB electrolyte preparation.				
c) L ⁻¹ in every unit refers to liter of vanadium solution.				

The process costs of the newly proposed catalytic production and conventional electrolysis were calculated as given in **Table R3, R4**. For the catalytic production, the process cost for 1 L of V^{3.5+} electrolyte is about \$ 0.0496, while that for the electrolysis production is \$ 0.0815. Therefore, 40% process cost reduction can be achieved with the catalytic production in comparison with the electrolysis due to the simplicity of the process. The calculation does not include labor cost, however, there will be a wider disparity between two processes when the labor cost is included. It should be noted that the other material costs (vanadium oxide and sulfuric acid) are identical between the two processes.

The discussions about comparison of catalytic production method and conventional electrolysis method are highly meaningful, thus all the relevant data were added to the revised manuscript and supplementary information as shown in the following.

Revised manuscript:

(Line 53-62)

Therefore, instead of chemical reduction, electrolysis of V⁴⁺ electrolyte has been employed using a VRFB stack^{17,19,20}. As indicated by the inventions from Skyllas-Kazacos' group, the reduction of V⁴⁺ electrolyte at the negative electrode can be coupled with either oxidation of V⁴⁺ at the positive electrode or water splitting reaction^{16,21}. Preparing V^{3.5+} electrolyte by using V⁴⁺ electrolyte in the positive electrode during electrolysis ensures impurity-free production, but the external reduction process for the surplus V^{4.5+} from the anode is required. The electrolysis method based on water splitting reaction can prevent the generation of surplus V^{4.5+} electrolyte. However, advanced engineering may be needed to address the vanadium ion

crossover to oxygen evolution reaction (OER) electrode and the carbon corrosion at OER electrode²².

(Line 341-350)

The catalytic process is highly cost-effective in terms of the process simplicity. The detailed cost analysis was conducted for the catalytic production and industrial electrolysis. For a comparison of capital cost with large-scale electrolysis using VRFB stack, the cost of a catalyst reactor with 40 L h⁻¹ production capacity is provided in **Supplementary note 2**. Since the vanadium materials used for the two processes are identical, the production cost comparison evaluated the reducing agent, feedstock, energy, and consumable materials costs. It was found that the newly proposed catalytic process could reduce the production cost by 40% compared to the industrial electrolysis process (**Supplementary note 2**) due to the simplicity of the process. It is believed that future optimization of the reactor technology can promise the further reduction of the production cost of VRFB electrolyte.

Revised supplementary information:

Supplementary note 2 Cost analysis for V^{3.5+} electrolyte production

Supplementary Fig. 6 Process diagrams for the production of 1.55 M V^{3.5+} electrolyte by (a) catalytic reaction, and (b) conventional electrolysis method.

The process diagram of the newly proposed process for V^{3.5+} electrolyte preparation using the catalytic reaction is given in **Supplementary Fig. 6a**. Starting from V⁴⁺ electrolyte, the V^{3.5+} electrolyte production can be achieved by a single step reaction using the catalytic reactor in a continuous manner. However, the conventional electrolysis method includes the external reduction step to fully utilize the feed vanadium solution (**Supplementary Fig. 6b**), therefore, this recycle stream does result in the need for larger equipment and some additional processing cost.

Supplementary Table 1 Cost of the catalyst reactor (Production capacity: 40 L h⁻¹)

	Material	Specification	Unit	Unit cost (\$)	Usage	Cost (\$)
Reactor	Teflon body	ϕ 300 * ϕ 220 * 300L	ea	625	2	1,250.0
	Teflon cap	ϕ 300*105	ea	419.2	2	838.3
	1/4" male connector		ea	2.9	2	5.8

	SUS bolt	M8*55	ea		30	16.7
	Cost of labor					166.7
	Sub total					2,277.5
Pt/C-decorated carbon felt	Pt/C	TEC10E50 (Tanaka)	g	20.5 ^{a)}	62	1,271.0
	Ionomer	D520 (DuPont)	L	316.7	0.72	228.0
	N-propyl alcohol		L	3.2	5.5	17.4
	Water		L	0.0033	4.5	0.015
	Carbon felt	GFD4.6 (SGL)	m ²	91.7	2.35	215.4
	Sub total					1,731.8
	Total					4,009.3

a) Tanaka TEC10E50: 2460YEN/g

For a fair comparison with the large-scale electrolysis production using VRFB stack, catalyst reactor with production capacity of 40 L h⁻¹ was considered. The large-scale reactor cost was analyzed as shown in **Supplementary Table 1**. The reactor mainly consists of a Teflon-based housing and Pt-decorated carbon felts. The catalytic activity of the reactor was measured to be 0.645 L g_{Pt/C}⁻¹h⁻¹, therefore, the required Pt/C catalyst for the reactor was 62 g (Pt: 28.8 g). Considering the amount of the catalyst and other components, the total cost of the reactor, which is capable of the production at 40 L h⁻¹, is \$ 4,009 (US dollar). One can compare the stack cost at the same production capacity of 40 L h⁻¹ to identify the capital cost difference.

Supplementary Table 2 Process cost for catalytic production of 1 L of 1.55 M V^{3.5+} electrolyte production

Component	Requirement	Unit	Unit cost	Cost (\$ L ⁻¹)
Formic acid	0.3875	mol L ⁻¹ ^{c)}	0.088 \$ mol ⁻¹	0.0341
Electrical heat input ^{a)}	55.14	Wh L ⁻¹	0.15 \$ kWh ⁻¹	0.0083
Consumable ^{b)}	┆	┆	┆	0.0072
Total				0.0496

a) Assumptions

- Heat capacity of electrolyte: 3.7 kJ kg⁻¹ °C⁻¹
- Heat recovery: 0.5 (fraction of total heat needed to raise temperature from 21°C to 95°C for reaction)
- Electricity cost: 0.15\$ kWh⁻¹ (The same as that for the electrolysis)

b) Assumptions: Pt/C coated electrode need to be replaced every 6000 h operation.

c) L⁻¹ in every unit refers to liter of vanadium solution.

Supplementary Table 3 Process cost for conventional method of 1 L of 1.55 M V^{3.5+} electrolyte production

Component	Requirement	Unit	Unit cost	Cost (\$ L ⁻¹)
Reducing agent ^{a)}	0.04	kg L ⁻¹ ^{c)}	1.2 \$ kg ⁻¹	0.0480
Electricity consumption	0.17	kWh L ⁻¹	0.15 \$ kWh ⁻¹	0.0255
Consumable ^{b)}	┆	┆	┆	0.0080
Total				0.0815

a) SO₂ was used as reducing agent.

b) Assumptions: Stack electrode need to be replaced every 500,000 L VRFB electrolyte preparation.

c) L⁻¹ in every unit refers to liter of vanadium solution.

The process costs of the newly proposed catalytic production and conventional electrolysis were calculated as given in **Supplementary Table 2, 3**. For the catalytic production, the process cost for 1 L of $V^{3.5+}$ electrolyte is about \$ 0.0496, while that for the electrolysis production is \$ 0.0815. Therefore, 40% process cost reduction can be achieved with the catalytic production in comparison with the electrolysis due to the simplicity of the process. The calculation does not include labor cost, however, there will be a wider disparity between two processes when the labor cost is included. It should be noted that the other material costs (vanadium oxide and sulfuric acid) are identical between the two processes.

The method described in this paper involves the use of a Pt-based catalyst to chemically reduce $V(4+)$ with formic acid. While Pt is shown to increase the kinetics of this reaction, a very important consideration has been overlooked. The authors claim that an impurity free electrolyte can be produced by this method, but the possibility of leaching Pt into the electrolyte during extended use of the catalyst has not been fully investigated over prolonged use under industrial conditions. As acknowledged by the authors, Pt and other noble metals are known to catalyze the $V(2+) + H^+$ reaction that displaces protons from solution, producing hydrogen gas. This reaction can occur even when the fully or partially charged battery is in standby mode and is a very undesirable reaction that needs to be avoided.

For the ICP-MS analysis, the electrolytes were diluted by a factor of 100 with DI-water. This dilution could have led to any leached Pt remain undetected.

Our response: We appreciate the reviewer for the comment on Pt dissolution. Noble metal impurity such as Pt ion can seriously impinge the cell performance as it catalyzes the hydrogen evolution reaction (HER) at the negative electrode (mostly during the charging step), therefore, Pt dissolution should not happen during the catalytic reaction. In this work, we provided ICP-MS data to prove that Pt does not leach into the electrolyte during the catalytic production of $V^{3.5+}$ electrolyte. However, the reviewer doubted about the dilution of electrolyte for the ICP-MS analysis, and surmised the dilution is the reason for the undetected Pt ion. However, for a proper and accurate ICP-MS analysis, the amount of total dissolved solids (TDS) is controlled to be below 0.5%. Therefore, the dilution is indispensable step for the ICP-MS analysis, and does not negate the accuracy of ICP-MS analysis.

Fig. R3 Particle-size-dependent potential-pH diagram for Pt / Pt²⁺ ([Pt²⁺] = 10⁻⁶ mol dm⁻³). The highlighted blue area represents the region of Pt²⁺ stability for 3 nm diameter platinum nanoparticle. “Reprinted with permission from (*J. Am. Chem. Soc.* **132**, 11722-11726 (2010)). Copyright (2010) American Chemical Society.”

We’d like to emphasize that the stability of Pt nanoparticle (3 nm) in V⁴⁺/V³⁺ acidic electrolytes is thermodynamically feasible. The potential-pH diagram for Pt/Pt²⁺ for 3 nm Pt nanoparticle was previously reported as shown in **Fig. R3**. The blue box represents Pt²⁺ stability region for the 3 nm Pt nanoparticle where Pt dissolution is favored. In the pHs below 0 (vanadium electrolytes), Pt oxidation potential is around 0.86 V and 0.71 V for 3 nm and 1.5 nm particle, respectively, and thus Pt is stable in the form of metallic state below the potential. The equilibrium redox potential of Pt in the catalytic reactor is determined by the redox potential of V³⁺/V⁴⁺ redox pair. The standard redox potential of V³⁺/V⁴⁺ is 0.34 V, which is far below than the Pt oxidation potential. Even when [V³⁺]/[V⁴⁺]=[10⁻⁶M/1.55M], the equilibrium redox potential (0.706V) is lower than the Pt oxidation potential according to Nernst equation. Therefore, the stability of Pt in the catalytic reactor is not surprising but just feasible. We added this consideration in the revised manuscript to provide a theoretical basis for the Pt stability.

Revised manuscript:

(Line 228-234)

The stability of Pt nanoparticle is quite reasonable considering the potential-pH diagram of Pt nanoparticle⁵¹ (**Supplementary Fig. 3**). For 3 nm-sized Pt nanoparticle, the Pt dissolution potential below pH 0 is 0.860 V, therefore, Pt is stable in the form of metallic state if the equilibrium potential of Pt, which is determined by the redox potential of V³⁺/V⁴⁺ redox pair, is below 0.86 V. Since the standard redox potential of V³⁺/V⁴⁺ (0.340 V) is far lower than the Pt oxidation potential, the stability of Pt/C in the vanadium electrolytes is thermodynamically feasible.

Revised supplementary information:

Supplementary Fig. 3 Particle-size-dependent potential-pH diagram for Pt / Pt²⁺ ([Pt²⁺] = 10⁻⁶ mol dm⁻³). The highlighted blue area represents the region of Pt²⁺ stability for 3 nm diameter Pt nanoparticle¹. “Reprinted with permission from (*J. Am. Chem. Soc.* **132**, 11722-11726 (2010)). Copyright (2010) American Chemical Society.”

The charge-discharge tests were performed between the voltage range 0.8 to 1.65 V but depending on the cell resistance, this may represent a low state-of charge at the end of the

charge cycle. The impact of excess hydrogen evolution may not therefore be seen at lower SOCs. The authors need to provide typical voltage vs time charge/discharge curves and give the open-circuit potentials at the end of the charge cycle to demonstrate that high SOCs are being reached during these cycling tests.

Fig. R4 More detailed electrochemical properties of the reactor electrolyte: (a) Charge/discharge curves (voltage vs time) for the reactor electrolyte during the first 25 cycles. (b) Open circuit voltages of the charged cells for the reactor electrolyte and electrolysis electrolyte after completing the charging step. (c) Cell performances of the reactor and electrolysis electrolytes with a wider voltage range (0.7 V ~ 1.9 V).

Our response: We are grateful for the reviewer's comment. In the comment, we were asked to provide voltage vs time charge/discharge curves with open-circuit voltage (OCV) at the end of charge cycle to demonstrate high state-of-charge (SOC) is achieved during the test. **Fig. R4a** shows the voltage profile of the VRFB cell with the electrolyte prepared by the catalyst reactor. The sharp increase of the cell voltage at the end of each charging step represents that the cell was fully charged. From the OCV after the charge, SOC was quantified by using the following equation (*European Chemical Bulletin* **1**, 511-519 (2012) / Patent Application US 15/101,092 (2016)).

$$\text{Positive half-cell: } E^+ = E^{\circ+'} - \frac{RT}{F} \left(\ln \left[\frac{1-SOC}{SOC} \right] \right) \quad (\text{R1})$$

$$\text{Negative half-cell: } E^- = E^{\circ-'} - \frac{RT}{F} \left(\ln \left[\frac{SOC}{1-SOC} \right] \right) \quad (\text{R2})$$

$$\text{Full cell: } \Delta E = \Delta E^{\circ'} - \frac{2RT}{F} \left(\ln \left[\frac{1-SOC}{SOC} \right] \right) \quad (\text{R3})$$

($E^{\circ+'} = 1.182 \text{ V}$, $E^{\circ-'} = -0.207 \text{ V}$, (*European Chemical Bulletin* **1**, 511-519 (2012))

According to **Eq. R3** and the OCV data (**Fig. R4b**), the value for SOC at the end of charging is 98%, which means the electrolyte was fully charged at the initial charging step. The SOC can be also calculated by the observed charge capacity and theoretical capacity of the electrolyte. The SOC value from the charge capacity was also as high as 96%. Therefore, the reviewers concern on the low SOC can be resolved.

The OCV values at the end of charging step during the cycling were compared for the reactor electrolyte (electrolyte prepared by the catalytic reactor) and electrolysis electrolyte (electrolyte prepared by the conventional electrolysis method) in **Fig. R4b**. It is clear that the OCV values are nearly the same for the two electrolytes, verifying that no other electrochemical reaction took place during the cell cycling except the vanadium redox reaction.

To further confirm the absence of any side reactions by Pt, we performed cell cycling test of the reactor electrolyte and the electrolysis electrolyte with a wider voltage range (0.7 V ~ 1.9 V). The charge cut-off voltage of 1.9 V leads to a higher SOC of 99%. As seen in **Fig. R4c**, the electrolysis electrolyte and reactor electrolyte showed identical cell performances for the wider voltage range.

The solution potential vs time should also be monitored for several days for a fully charged solution to verify that no hydrogen ion displacement reaction is occurring that leads to hydrogen evolution on standing at high SOC.

Fig. R5 Self-discharge test for the fully charged cells with reactor and electrolysis electrolytes.

Our response: We thank the reviewer for the useful comments. Reviewer advised to conduct

self-discharge test after fully charging the solution as HER can take place even in the rest state when solution is in high SOC. **Fig. R5** shows self-discharge test result comparing electrolyte prepared by using catalytic reactor and electrolysis method. Electrolyte prepared by using catalyst reactor showed equivalent result with electrolyte produced by using electrolysis, again verifying that Pt ion is not included in the reactor electrolyte by the rationale from excluded HER. We think the discussion about Pt dissolution is meaningful to potential readers of our manuscript, thus newly acquired data were added to the revised manuscript and supplementary information as shown below.

Revised manuscript:

(Line 304-323)

Fig. 7 Electrochemical stability of the reactor electrolyte: (a) Cell performances of the electrolytes prepared by the catalyst reactor and electrolysis under a wider voltage window (0.7 V ~ 1.9 V). (b) Self-discharge test for the reactor and electrolysis electrolytes after the charge up to 2.4 V.

To achieve high VRFB performances, the stability of Pt catalyst during the catalytic production is highly important because the dissolved Pt ion can cause hydrogen evolution at the negative

electrode, lowering efficiency and cycling stability. Since the reduction of Pt^{2+} to Pt and subsequent hydrogen evolution are accelerated with increasing SOC, the cells were operated in a wider voltage window of 0.7 ~ 1.9 V to further confirm the absence of the Pt dissolution. The charge cut-off voltage of 1.9 V leads to a higher SOC of 99% (**Supplementary note 1**). As it is shown in **Fig. 7a**, the reactor electrolyte showed identical efficiencies with the electrolysis electrolyte without any discrepancies even at the high SOC condition, proving the absence of any side reactions. Since HER can take place even in the rest stage when solution is in high SOC, self-discharge test was also conducted to garner consolidated conclusion about the high stability of Pt catalyst in the reactor. The reactor electrolyte manifested negligible differences with the electrolysis electrolyte in the self-discharge test after the charge up to 2.4 V (**Fig. 7b**), again verifying that Pt ion is not included in the reactor electrolyte by the rationale from excluded HER.

Revised supplementary information:

Supplementary note 1 Calculation of SOC

SOC of initially charged electrolyte was calculated using below equation^{2,3}.

$$\text{Positive half-cell: } E^+ = E^{o+'} - \frac{RT}{F} \left(\ln \left[\frac{1-\text{SOC}}{\text{SOC}} \right] \right) \quad (\text{S1})$$

$$\text{Negative half-cell: } E^- = E^{o-' } - \frac{RT}{F} \left(\ln \left[\frac{\text{SOC}}{1-\text{SOC}} \right] \right) \quad (\text{S2})$$

$$\text{Full cell: } \Delta E = \Delta E^{o'} - \frac{2RT}{F} \left(\ln \left[\frac{1-\text{SOC}}{\text{SOC}} \right] \right) \quad (\text{S3})$$

In the previous work of Kazacos's group, formal potentials for vanadium electrolyte of 1.6 M vanadium and sulfate level in 4 M ~ 4.2 M were calculated; $E^{o+'} = 1.182$ V for positive half-cell and $E^{o-' } = -0.207$ V for negative half-cell, respectively. Since the composition of reactor electrolyte (1.55 M of vanadium ion + 4.17 M sulfate ion) is almost identical to the electrolyte in previous work, these values were used for calculating the SOC of the reactor electrolyte after initial charging step. By using **Eq. S3**, the SOC of initially charged electrolyte was about 99% which means the electrolyte was fully charged when it was charged up to 1.9 V. SOC calculated from charge capacity exhibited same value (99%) consolidating the fact that high SOC was achieved by charging the electrolyte to 1.9 V. SOC of electrolyte charged up to 1.65 V also showed high SOC (98%), verifying that 0.8 V ~ 1.65 V is wide enough voltage range to check HER reaction.

Before this paper can be accepted for publication in Nature Communications therefore, more

information should be provided as detailed above.

Our response: We firmly believe every tests conducted manifest coherent result that Pt dissolution is not involved in our suggested method, and the conclusion about the stability of Pt is in accord with the previous reports in the field of phosphoric acid fuel cell (*J. Power Sources*, **164**, 126-133 (2007)/ *Electrochemical cells*, **127**, 1219-1224 (1980)) where Pt catalyst is utilized in an extreme condition (85% H₃PO₄ electrolyte, operating above 150 °C) without showing failure by dissolution. Therefore, we suggest that the catalytic production of V^{3.5+} electrolyte can certainly be a new path for preparing VRFB electrolyte.

Once again, we highly appreciate every useful comments and advises, these truly helped us to improve the quality of manuscript.

Reviewer #2 (Remarks to the Author):

The cost of producing the vanadium-based electrolyte for redox flow batteries is one of the main causes of the still higher storage costs compared to lithium ion batteries, for example. The proposed method could certainly contribute to the required cost reduction. However, in order to be able to evaluate this, at least a cost estimate should be made in comparison to the electrolysis method, taking into account the much more complex reactor and the high costs for the catalyst used.

Our response: We highly appreciate for the reviewer's useful comment on our manuscript. We also agree about the necessity of cost analysis in our work, therefore, cost comparisons of newly proposed catalytic production method and conventional electrolysis method are done as shown below.

Fig. R2 Process diagrams for the production of 1.55 M V^{3.5+} electrolyte by (a) catalytic reaction, and (b) conventional electrolysis method.

The process diagram of the newly proposed process for V^{3.5+} electrolyte preparation using the catalytic reaction is given in **Fig. R2a**. Starting from V⁴⁺ electrolyte, the V^{3.5+} electrolyte

production can be achieved by a single step reaction using the catalytic reactor in a continuous manner. The conventional process modelled used the flow chart of **Fig. R2b**. It uses chemical reduction of V^{5+} to V^{4+} with SO_2 , followed by electrolytic reduction to $V^{3.5+}$ finished product. Note that in this conventional process, it is possible to avoid the higher capital costs of electrolyzers using the OER counter electrode described in the reference (Patent Application US 10/226,751 (2006)). Instead, a conventional VRFB cell stack with carbon felt electrodes can be used for both anolyte and catholyte. The counter-electrode is fed with the same V^{4+} material which is converted to $V^{4.5+}$ at the counter electrode. The $V^{4.5+}$ stream is then recycled to the front of process, reduced to V^{4+} and reprocessed. This recycle stream does result in the need for larger equipment and some additional processing cost.

Table R2 Cost of the catalyst reactor (Production capacity: 40 L h⁻¹)

	Material	Specification	Unit	Unit cost (\$)	Usage	Cost (\$)
Reactor	Teflon body	ϕ 300 * ϕ 220 * 300L	ea	625	2	1,250.0
	Teflon cap	ϕ 300*105	ea	419.2	2	838.3
	1/4" male connector		ea	2.9	2	5.8
	SUS bolt	M8*55	ea		30	16.7
	Cost of labor					166.7
Sub total						2,277.5
Pt/C-decorated carbon felt	Pt/C	TEC10E50 (Tanaka)	g	20.5 ^{a)}	62	1,271.0
	Ionomer	D520 (DuPont)	L	316.7	0.72	228.0
	N-propyl alcohol		L	3.2	5.5	17.4
	Water		L	0.0033	4.5	0.015
	Carbon felt	GFD4.6 (SGL)	m ²	91.7	2.35	215.4
Sub total						1,731.8
Total						4,009.3

a) Tanaka TEC10E50: 2460YEN/g

For a fair comparison with the large-scale electrolysis production using VRFB stack, catalyst reactor with production capacity of 40 L h⁻¹ was derived by the experimental results we obtained from the lab-scale prototype reactor. The large-scale reactor cost was analyzed as shown in **Table R2**. The reactor mainly consists of a Teflon-based housing and Pt-decorated carbon felts. The catalytic activity of the reactor was measured to be 0.645 L g_{Pt/C}⁻¹ h⁻¹, therefore, the required Pt/C catalyst for the reactor was 62 g (Pt: 28.8 g). Considering the amount of the catalyst and other components, the total cost of the reactor, which is capable of the production at 40 L h⁻¹, is \$ 4,009 (US dollar). One can compare the stack cost at the same production capacity of 40 L h⁻¹ to identify the capital cost difference.

Table R3 Process cost for the catalytic production of 1 L of 1.55 M $V^{3.5+}$ electrolyte

Component	Requirement	Unit	Unit cost	Cost (\$ L ⁻¹)
Formic acid	0.3875	mol L ^{-1 c)}	0.088 \$ mol ⁻¹	0.0341
Electrical heat input ^{a)}	55.14	Wh L ⁻¹	0.15 \$ kWh ⁻¹	0.0083

Consumable ^{b)}	-	-	-	0.0072
Total				0.0496

- a) Assumptions
1. Heat capacity of electrolyte: 3.7 kJ kg⁻¹ °C⁻¹
 2. Heat recovery: 0.5 (fraction of total heat needed to raise temperature from 21°C to 95°C for reaction)
 3. Electricity cost: 0.15\$ kWh⁻¹ (The same as that for the electrolysis)
- b) Assumptions: Pt/C coated electrode need to be replaced every 6000 h operation.
- c) L⁻¹ in every unit refers to liter of vanadium solution.

Table R4 Process cost for the conventional electrolysis production of 1 L of 1.55 M V^{3.5+} electrolyte

Component	Requirement	Unit	Unit cost	Cost (\$ L ⁻¹)
Reducing agent ^{a)}	0.04	kg L ^{-1 c)}	1.2 \$ kg ⁻¹	0.0480
Electricity consumption	0.17	kWh L ⁻¹	0.15 \$ kWh ⁻¹	0.0255
Consumable ^{b)}	-	-	-	0.0080
Total				0.0815

- a) SO₂ was used as reducing agent.
- b) Assumptions: Stack electrode need to be replaced every 500,000 L VRFB electrolyte preparation.
- c) L⁻¹ in every unit refers to liter of vanadium solution.

The process costs of the newly proposed catalytic production and conventional electrolysis were calculated as given in **Table R3, R4**. For the catalytic production, the process cost for 1 L of V^{3.5+} electrolyte is about \$ 0.0496, while that for the electrolysis production is \$ 0.0815. Therefore, 40% process cost reduction can be achieved with the catalytic production in comparison with the electrolysis due to the simplicity of the process. The calculation does not include labor cost, however, there will be a wider disparity between two processes when the labor cost is included. It should be noted that the other material costs (vanadium oxide and sulfuric acid) are identical between the two processes.

The discussions about comparison of catalytic production method and conventional electrolysis method are highly meaningful, thus all the relevant data were added to the revised manuscript and supplementary information as shown in the following.

Revised manuscript:

(Line 53-62)

Therefore, instead of chemical reduction, electrolysis of V⁴⁺ electrolyte has been employed using a VRFB stack^{17,19,20}. As indicated by the inventions from Skyllas-Kazacos' group, the reduction of V⁴⁺ electrolyte at the negative electrode can be coupled with either oxidation of V⁴⁺ at the positive electrode or water splitting reaction^{16,21}. Preparing V^{3.5+} electrolyte by using V⁴⁺ electrolyte in the positive electrode during electrolysis ensures impurity-free production, but the external reduction process for the surplus V^{4.5+} from the anode is required. The electrolysis method based on water splitting reaction can prevent the generation of surplus V^{4.5+}

electrolyte. However, advanced engineering may be needed to address the vanadium ion crossover to oxygen evolution reaction (OER) electrode and the carbon corrosion at OER electrode²².

(Line 341-350)

The catalytic process is highly cost-effective in terms of the process simplicity. The detailed cost analysis was conducted for the catalytic production and industrial electrolysis. For a comparison of capital cost with large-scale electrolysis using VRFB stack, the cost of a catalyst reactor with 40 L h⁻¹ production capacity is provided in **Supplementary note 2**. Since the vanadium materials used for the two processes are identical, the production cost comparison evaluated the reducing agent, feedstock, energy, and consumable materials costs. It was found that the newly proposed catalytic process could reduce the production cost by 40% compared to the industrial electrolysis process (**Supplementary note 2**) due to the simplicity of the process. It is believed that future optimization of the reactor technology can promise the further reduction of the production cost of VRFB electrolyte.

Revised supplementary information:

Supplementary note 2 Cost analysis for V^{3.5+} electrolyte production

Supplementary Fig. 6 Process diagrams for the production of 1.55 M V^{3.5+} electrolyte by (a) catalytic reaction, and (b) conventional electrolysis method.

The process diagram of the newly proposed process for V^{3.5+} electrolyte preparation using the catalytic reaction is given in **Supplementary Fig. 6a**. Starting from V⁴⁺ electrolyte, the V^{3.5+} electrolyte production can be achieved by a single step reaction using the catalytic reactor in a continuous manner. However, the conventional electrolysis method includes the external reduction step to fully utilize the feed vanadium solution (**Supplementary Fig. 6b**), therefore, this recycle stream does result in the need for larger equipment and some additional processing cost.

Supplementary Table 1 Cost of the catalyst reactor (Production capacity: 40 L h⁻¹)

	Material	Specification	Unit	Unit cost (\$)	Usage	Cost (\$)
Reactor	Teflon body	¢ 300 * ¢ 220 * 300L	ea	625	2	1,250.0

	Teflon cap	φ 300*105	ea	419.2	2	838.3
	1/4" male connector		ea	2.9	2	5.8
	SUS bolt	M8*55	ea		30	16.7
	Cost of labor					166.7
	Sub total					2,277.5
Pt/C-decorated carbon felt	Pt/C	TEC10E50 (Tanaka)	g	20.5 ^{a)}	62	1,271.0
	Ionomer	D520 (DuPont)	L	316.7	0.72	228.0
	N-propyl alcohol		L	3.2	5.5	17.4
	Water		L	0.0033	4.5	0.015
	Carbon felt	GFD4.6 (SGL)	m ²	91.7	2.35	215.4
	Sub total					1,731.8
	Total					4,009.3

a) Tanaka TEC10E50: 2460YEN/g

For a fair comparison with the large-scale electrolysis production using VRFB stack, catalyst reactor with production capacity of 40 L h⁻¹ was considered. The large-scale reactor cost was analyzed as shown in **Supplementary Table 1**. The reactor mainly consists of a Teflon-based housing and Pt-decorated carbon felts. The catalytic activity of the reactor was measured to be 0.645 L g_{Pt/C}⁻¹h⁻¹, therefore, the required Pt/C catalyst for the reactor was 62 g (Pt: 28.8 g). Considering the amount of the catalyst and other components, the total cost of the reactor, which is capable of the production at 40 L h⁻¹, is \$ 4,009 (US dollar). One can compare the stack cost at the same production capacity of 40 L h⁻¹ to identify the capital cost difference.

Supplementary Table 2 Process cost for catalytic production of 1 L of 1.55 M V^{3.5+} electrolyte production

Component	Requirement	Unit	Unit cost	Cost (\$ L ⁻¹)
Formic acid	0.3875	mol L ⁻¹ ^{c)}	0.088 \$ mol ⁻¹	0.0341
Electrical heat input ^{a)}	55.14	Wh L ⁻¹	0.15 \$ kWh ⁻¹	0.0083
Consumable ^{b)}	┆	┆	┆	0.0072
Total				0.0496

a) Assumptions

- Heat capacity of electrolyte: 3.7 kJ kg⁻¹ °C⁻¹
- Heat recovery: 0.5 (fraction of total heat needed to raise temperature from 21°C to 95°C for reaction)
- Electricity cost: 0.15\$ kWh⁻¹ (The same as that for the electrolysis)

b) Assumptions: Pt/C coated electrode need to be replaced every 6000 h operation.

c) L⁻¹ in every unit refers to liter of vanadium solution.

Supplementary Table 3 Process cost for conventional method of 1 L of 1.55 M V^{3.5+} electrolyte production

Component	Requirement	Unit	Unit cost	Cost (\$ L ⁻¹)
Reducing agent ^{a)}	0.04	kg L ⁻¹ ^{c)}	1.2 \$ kg ⁻¹	0.0480
Electricity consumption	0.17	kWh L ⁻¹	0.15 \$ kWh ⁻¹	0.0255
Consumable ^{b)}	┆	┆	┆	0.0080
Total				0.0815

- a) SO₂ was used as reducing agent.
- b) Assumptions: Stack electrode need to be replaced every 500,000 L VRFB electrolyte preparation.
- c) L⁻¹ in every unit refers to liter of vanadium solution.

The process costs of the newly proposed catalytic production and conventional electrolysis were calculated as given in **Supplementary Table 2, 3**. For the catalytic production, the process cost for 1 L of V^{3.5+} electrolyte is about \$ 0.0496, while that for the electrolysis production is \$ 0.0815. Therefore, 40% process cost reduction can be achieved with the catalytic production in comparison with the electrolysis due to the simplicity of the process. The calculation does not include labor cost, however, there will be a wider disparity between two processes when the labor cost is included. It should be noted that the other material costs (vanadium oxide and sulfuric acid) are identical between the two processes.

Minor changes:

Line 176: The heterogeneous (remove: catalysts) Pd- and Pt-based catalysts are most commonly used because 176 of their high activity.

Our response: We thank the reviewer for careful review. As it is commented, we made the change in the manuscript as below.

Revised manuscript:

(Line 177-178)

The heterogeneous Pd- and Pt-based catalysts are most commonly used because of their high activity.

Reviewer #3 (Remarks to the Author):

The authors propose a simpler method for chemical production of impurity-free V^{3.5+} electrolyte by utilizing organic fuel cell catalysts. The new method features the use of catalyst and an organic reducing agent (ORA), which enable the reduction of V⁴⁺ to V³⁺ without releasing any undesired impurities. However, the major bottleneck for all vanadium redox flow battery is the materials cost of vanadium itself, in addition, this method uses Pt/C in the process, which is very expensive (precious metals). I do not see economical advantage of using this method. The authors should provide more in-depth analysis to justify that this method will address the high cost of Vanadium electrolyte. For example, how much cost reduction can be achieved by using the proposed method? Is precious metal required in this process? If so, what's the implication for price? From my understanding, the vanadium redox flow battery using

conventional method can perform as good as what is reported in the manuscript if not better. Therefore, I do not find strong justifications to support the publication of this manuscript at its present form.

Our response: We highly appreciate for the reviewer’s useful comment on our manuscript. We also agree about the necessity of cost analysis in our work, therefore, cost comparisons of newly proposed catalytic production method and conventional electrolysis method are done as shown below. Also, we would like to emphasize that our current work focuses on the development of a simpler route to produce $V^{3.5+}$ electrolyte using the catalytic reaction without releasing any impurities. Both the catalytic reaction and conventional electrolysis methods do not generate any impurities during the electrolyte production, therefore, the electrochemical performances of the two electrolytes do not differ. However, the newly developed process is much simpler than the conventional electrolysis, which reduces the production cost as described in the following.

Fig. R2 Process diagrams for the production of 1.55 M $V^{3.5+}$ electrolyte by (a) catalytic reaction, and (b) conventional electrolysis method.

The process diagram of the newly proposed process for $V^{3.5+}$ electrolyte preparation using the catalytic reaction is given in **Fig. R2a**. Starting from V^{4+} electrolyte, the $V^{3.5+}$ electrolyte production can be achieved by a single step reaction using the catalytic reactor in a continuous manner. The conventional process modelled used the flow chart of **Fig. R2b**. It uses chemical reduction of V^{5+} to V^{4+} with SO_2 , followed by electrolytic reduction to $V^{3.5+}$ finished product. Note that in this conventional process, it is possible to avoid the higher capital costs of electrolyzers using the OER counter electrode described in the reference (Patent Application US 10/226,751 (2006)). Instead, a conventional VRFB cell stack with carbon felt electrodes can be used for both anolyte and catholyte. The counter-electrode is fed with the same V^{4+} material which is converted to $V^{4.5+}$ at the counter electrode. The $V^{4.5+}$ stream is then recycled to the front of process, reduced to V^{4+} and reprocessed. This recycle stream does result in the need for larger equipment and some additional processing cost.

Table R2 Cost of the catalyst reactor (Production capacity: 40 L h⁻¹)

Material	Specification	Unit	Unit cost (\$)	Usage	Cost (\$)
----------	---------------	------	----------------	-------	-----------

Reactor	Teflon body	φ 300 * φ 220 * 300L	ea	625	2	1,250.0
	Teflon cap	φ 300*105	ea	419.2	2	838.3
	1/4" male connector		ea	2.9	2	5.8
	SUS bolt	M8*55	ea		30	16.7
	Cost of labor					166.7
Sub total						2,277.5
Pt/C-decorated carbon felt	Pt/C	TEC10E50 (Tanaka)	g	20.5 ^{a)}	62	1,271.0
	Ionomer	D520 (DuPont)	L	316.7	0.72	228.0
	N-propyl alcohol		L	3.2	5.5	17.4
	Water		L	0.0033	4.5	0.015
	Carbon felt	GFD4.6 (SGL)	m ²	91.7	2.35	215.4
Sub total						1,731.8
Total						4,009.3

a) Tanaka TEC10E50: 2460YEN/g

For a fair comparison with the large-scale electrolysis production using VRFB stack, catalyst reactor with production capacity of 40 L h⁻¹ was derived by the experimental results we obtained from the lab-scale prototype reactor. The large-scale reactor cost was analyzed as shown in **Table R2**. The reactor mainly consists of a Teflon-based housing and Pt-decorated carbon felts. The catalytic activity of the reactor was measured to be 0.645 L g_{Pt/C}⁻¹ h⁻¹, therefore, the required Pt/C catalyst for the reactor was 62 g (Pt: 28.8 g). Considering the amount of the catalyst and other components, the total cost of the reactor, which is capable of the production at 40 L h⁻¹, is \$ 4,009 (US dollar). One can compare the stack cost at the same production capacity of 40 L h⁻¹ to identify the capital cost difference.

Table R3 Process cost for the catalytic production of 1 L of 1.55 M V^{3.5+} electrolyte

Component	Requirement	Unit	Unit cost	Cost (\$ L ⁻¹)
Formic acid	0.3875	mol L ^{-1 c)}	0.088 \$ mol ⁻¹	0.0341
Electrical heat input ^{a)}	55.14	Wh L ⁻¹	0.15 \$ kWh ⁻¹	0.0083
Consumable ^{b)}	-	-	-	0.0072
Total				0.0496

a) Assumptions

1. Heat capacity of electrolyte: 3.7 kJ kg⁻¹ °C⁻¹
2. Heat recovery: 0.5 (fraction of total heat needed to raise temperature from 21°C to 95°C for reaction)
3. Electricity cost: 0.15\$ kWh⁻¹ (The same as that for the electrolysis)

b) Assumptions: Pt/C coated electrode need to be replaced every 6000 h operation.

c) L⁻¹ in every unit refers to liter of vanadium solution.

Table R4 Process cost for the conventional electrolysis production of 1 L of 1.55 M V^{3.5+} electrolyte

Component	Requirement	Unit	Unit cost	Cost (\$ L ⁻¹)
Reducing agent ^{a)}	0.04	kg L ^{-1 c)}	1.2 \$ kg ⁻¹	0.0480
Electricity consumption	0.17	kWh L ⁻¹	0.15 \$ kWh ⁻¹	0.0255

Consumable ^{b)}	-	-	-	0.0080
Total				0.0815

- a) SO₂ was used as reducing agent.
b) Assumptions: Stack electrode need to be replaced every 500,000 L VRFB electrolyte preparation.
c) L⁻¹ in every unit refers to liter of vanadium solution.

The process costs of the newly proposed catalytic production and conventional electrolysis were calculated as given in **Table R3, R4**. For the catalytic production, the process cost for 1 L of V^{3.5+} electrolyte is about \$ 0.0496, while that for the electrolysis production is \$ 0.0815. Therefore, 40% process cost reduction can be achieved with the catalytic production in comparison with the electrolysis due to the simplicity of the process. The calculation does not include labor cost, however, there will be a wider disparity between two processes when the labor cost is included. It should be noted that the other material costs (vanadium oxide and sulfuric acid) are identical between the two processes.

The discussions about comparison of catalytic production method and conventional electrolysis method are highly meaningful, thus all the relevant data were added to the revised manuscript and supplementary information as shown in the following.

Revised manuscript:

(Line 53-62)

Therefore, instead of chemical reduction, electrolysis of V⁴⁺ electrolyte has been employed using a VRFB stack^{17,19,20}. As indicated by the inventions from Skyllas-Kazacos' group, the reduction of V⁴⁺ electrolyte at the negative electrode can be coupled with either oxidation of V⁴⁺ at the positive electrode or water splitting reaction^{16,21}. Preparing V^{3.5+} electrolyte by using V⁴⁺ electrolyte in the positive electrode during electrolysis ensures impurity-free production, but the external reduction process for the surplus V^{4.5+} from the anode is required. The electrolysis method based on water splitting reaction can prevent the generation of surplus V^{4.5+} electrolyte. However, advanced engineering may be needed to address the vanadium ion crossover to oxygen evolution reaction (OER) electrode and the carbon corrosion at OER electrode²².

(Line 341-350)

The catalytic process is highly cost-effective in terms of the process simplicity. The detailed cost analysis was conducted for the catalytic production and industrial electrolysis. For a comparison of capital cost with large-scale electrolysis using VRFB stack, the cost of a catalyst reactor with 40 L h⁻¹ production capacity is provided in **Supplementary note 2**. Since the vanadium materials used for the two processes are identical, the production cost comparison evaluated the reducing agent, feedstock, energy, and consumable materials costs. It was found that the newly proposed catalytic process could reduce the production cost by 40% compared

to the industrial electrolysis process (**Supplementary note 2**) due to the simplicity of the process. It is believed that future optimization of the reactor technology can promise the further reduction of the production cost of VRFB electrolyte.

Revised supplementary information:

Supplementary note 2 Cost analysis for $V^{3.5+}$ electrolyte production

Supplementary Fig. 6 Process diagrams for the production of 1.55 M $V^{3.5+}$ electrolyte by (a) catalytic reaction, and (b) conventional electrolysis method.

The process diagram of the newly proposed process for $V^{3.5+}$ electrolyte preparation using the catalytic reaction is given in **Supplementary Fig. 6a**. Starting from V^{4+} electrolyte, the $V^{3.5+}$ electrolyte production can be achieved by a single step reaction using the catalytic reactor in a continuous manner. However, the conventional electrolysis method includes the external reduction step to fully utilize the feed vanadium solution (**Supplementary Fig. 6b**), therefore, this recycle stream does result in the need for larger equipment and some additional processing cost.

Supplementary Table 1 Cost of the catalyst reactor (Production capacity: 40 L h⁻¹)

	Material	Specification	Unit	Unit cost (\$)	Usage	Cost (\$)
Reactor	Teflon body	ϕ 300 * ϕ 220 * 300L	ea	625	2	1,250.0
	Teflon cap	ϕ 300*105	ea	419.2	2	838.3
	1/4" male connector		ea	2.9	2	5.8
	SUS bolt	M8*55	ea		30	16.7
	Cost of labor					166.7
	Sub total					2,277.5
Pt/C-decorated carbon felt	Pt/C	TEC10E50 (Tanaka)	g	20.5 ^{a)}	62	1,271.0
	Ionomer	D520 (DuPont)	L	316.7	0.72	228.0
	N-propyl alcohol		L	3.2	5.5	17.4
	Water		L	0.0033	4.5	0.015
	Carbon felt	GFD4.6 (SGL)	m ²	91.7	2.35	215.4
	Sub total					1,731.8
	Total					4,009.3

a) Tanaka TEC10E50: 2460YEN/g

For a fair comparison with the large-scale electrolysis production using VRFB stack, catalyst reactor with production capacity of 40 L h⁻¹ was considered. The large-scale reactor cost was

analyzed as shown in **Supplementary Table 1**. The reactor mainly consists of a Teflon-based housing and Pt-decorated carbon felts. The catalytic activity of the reactor was measured to be $0.645 \text{ L } g_{\text{Pt/C}}^{-1} h^{-1}$, therefore, the required Pt/C catalyst for the reactor was 62 g (Pt: 28.8 g). Considering the amount of the catalyst and other components, the total cost of the reactor, which is capable of the production at 40 L h^{-1} , is \$ 4,009 (US dollar). One can compare the stack cost at the same production capacity of 40 L h^{-1} to identify the capital cost difference.

Supplementary Table 2 Process cost for catalytic production of 1 L of $1.55 \text{ M V}^{3.5+}$ electrolyte production

Component	Requirement	Unit	Unit cost	Cost (\$ L ⁻¹)
Formic acid	0.3875	mol L ^{-1 c)}	0.088 \$ mol ⁻¹	0.0341
Electrical heat input ^{a)}	55.14	Wh L ⁻¹	0.15 \$ kWh ⁻¹	0.0083
Consumable ^{b)}	–	–	–	0.0072
Total				0.0496

a) Assumptions

1. Heat capacity of electrolyte: $3.7 \text{ kJ kg}^{-1} \text{ } ^\circ\text{C}^{-1}$
2. Heat recovery: 0.5 (fraction of total heat needed to raise temperature from 21°C to 95°C for reaction)
3. Electricity cost: $0.15 \text{ $ kWh}^{-1}$ (The same as that for the electrolysis)

b) Assumptions: Pt/C coated electrode need to be replaced every 6000 h operation.

c) L⁻¹ in every unit refers to liter of vanadium solution.

Supplementary Table 3 Process cost for conventional method of 1 L of $1.55 \text{ M V}^{3.5+}$ electrolyte production

Component	Requirement	Unit	Unit cost	Cost (\$ L ⁻¹)
Reducing agent ^{a)}	0.04	kg L ^{-1 c)}	1.2 \$ kg ⁻¹	0.0480
Electricity consumption	0.17	kWh L ⁻¹	0.15 \$ kWh ⁻¹	0.0255
Consumable ^{b)}	–	–	–	0.0080
Total				0.0815

a) SO₂ was used as reducing agent.

b) Assumptions: Stack electrode need to be replaced every 500,000 L VRFB electrolyte preparation.

c) L⁻¹ in every unit refers to liter of vanadium solution.

The process costs of the newly proposed catalytic production and conventional electrolysis were calculated as given in **Supplementary Table 2, 3**. For the catalytic production, the process cost for 1 L of $\text{V}^{3.5+}$ electrolyte is about \$ 0.0496, while that for the electrolysis production is \$ 0.0815. Therefore, 40% process cost reduction can be achieved with the catalytic production in comparison with the electrolysis due to the simplicity of the process. The calculation does not include labor cost, however, there will be a wider disparity between two processes when the labor cost is included. It should be noted that the other material costs (vanadium oxide and sulfuric acid) are identical between the two processes.

REVIEWERS' COMMENTS:

Reviewer #1 (Remarks to the Author):

I am satisfied that the authors have adequately addressed the comments in my original review, so I am happy to recommend publication of this manuscript.

Reviewer #2 (Remarks to the Author):

Referring to my first review, the authors dealt very intensively with a cost estimate for the considered processes, which was the main point of criticism.

They calculated the process costs of the newly proposed catalytic production and the conventional electrolysis and demonstrated a 40% process cost reduction with the catalytic production in comparison with the electrolysis due to the simplicity of the process. This highly improved the quality and the meaningfulness of the paper.

Reviewer #3 (Remarks to the Author):

The authors have developed comprehensive cost estimation model to address my questions. Although at the end the cost reduction is not that much, there is still some improvement. I am still concerned about the use of Pt because it is precious metal and its price will continue go up since fuel cells also need Pt. Nevertheless, I appreciate the authors effort and think it is an interesting study.